# SelfXit: An Unsupervised Early Exit Mechanism for Deep Neural Networks

**Hossein KhademSohi**                                    *hossein.khademsohi@ucalgary.ca*
**Mohammadamin Abedi**                              *mohammadamin.abedi@ucalgary.ca*
**Yani Ioannou**                                                  *yani.ioannou@ucalgary.ca*
**Steve Drew**                                                      *steve.drew@ucalgary.ca*
*Department of Electrical and Software Engineering*
*University of Calgary*
*Calgary, AB, Canada*

**Pooyan Jamshidi**                                              *pjamshid@cse.sc.edu*
*Department of Computing Science and Engineering*
*University of South Carolina*
*Columbia, SC, USA*

**Hadi Hemmati**                                                  *hemmati@yorku.ca*
*Department of Electrical Engineering and Computer Science*
*University of York*
*Toronto, ON, Canada*

**Reviewed on OpenReview:** *https://openreview.net/forum?id=h4rUKKfl5S*

## Abstract

Deep Neural Networks (DNNs) have become an essential component in many application domains, including web-based services. A variety of these services require high throughput and (close to) real-time features, for instance, to respond or react to users' requests or to process a stream of incoming data on time. However, the trend in DNN design is towards larger models with many layers and parameters to achieve more accurate results. Although these models are often pre-trained, the computational complexity in such large models can still be relatively significant, hindering low inference latency. In this paper, we propose SelfXit, an end-to-end automated early exiting solution to improve the performance of DNN-based vision services in terms of computational complexity and inference latency. SelfXit adopts the ideas of self-distillation of DNN models and early exits specifically for vision applications. The proposed solution is an automated unsupervised early exiting mechanism that allows early exiting of a large model during inference time if the early exit model in one of the early exits is confident enough for final prediction. One of the main contributions of this paper is that we have implemented the idea as an unsupervised early exiting, meaning that the early exit models do not need access to training data and perform solely based on the incoming data at run-time, making it suitable for applications using pre-trained models. The results of our experiments on two vision tasks (image classification and object detection) show that, on average, early exiting can reduce the computational complexity of these services up to 58% (in terms of FLOP count) and improve their inference latency up to 46% with a low to zero reduction in accuracy. SelfXit also outperforms existing methods, particularly on complex models and larger datasets. It achieves a notable reduction in latency of 51.6% and 30.4% on CIFAR100/Resnet50, with an accompanying increase in accuracy of 2.31% and 0.72%, on average, compared to GATI and BranchyNet. We released code and replication package at https://github.com/hoseinkhs/AutoCacheLayer/.

# 1 Introduction

Deep Neural Networks (DNNs) are incorporated in real-world applications used by a wide spectrum of industry sectors including healthcare (Shorten et al., 2021; Fink et al., 2020), finance (Huang et al., 2020; Culkin & Das, 2017), self-driving vehicles (Swinney & Woods, 2021), and cybersecurity (Ferrag et al., 2020). These applications utilize DNNs in various fields such as computer vision (Hassaballah & Awad, 2020; Swinney & Woods, 2021), audio signal processing (Arakawa et al., 2019; Tashev & Mirsamadi, 2016),and natural language processing (Otter et al., 2021). Many services in large companies such as Google and Amazon have DNN-based back-end software (e.g., Google Lens and Amazon Rekognition) with tremendous volume of queries per second. For instance, Google processes over 99,000 searches every second (Mohsin, 2022) and spends a substantial amount of computation power and time on the run time of their models (Xiang & Kim, 2019). These services are often time-sensitive and resource-intensive and require high availability and reliability.

Now the question is how fast the current state-of-the-art (SOTA) DNN models are at inference time and to what extent they can provide low-latency responses to queries. The SOTA model depends on the application domain and the problem at hand. However, the trend in DNN design is indeed toward pre-trained large-scale models due to their reduced training cost (only fine-tuning) while providing dominating results (since they are huge models trained on an extensive dataset).

One of the downsides of large-scale models (pre-trained or not) is their high inference latency. Although the inference latency is usually negligible per instance, as discussed, a relatively slow inference can jeopardize a service's performance in terms of throughput when the QPS is high.

In general, in a DNN-based software development and deployment pipeline, the inference stage is part of the so-called "model serving" process, which enables the model to serve inference requests or jobs (Xiang & Kim, 2019) by directly loading the model in the process or by employing serving frameworks such as TensorFlow Serving (Olston et al., 2017) or Clipper (Crankshaw et al., 2017).

The inference phase is an expensive stage in the life cycle of a deep neural model in terms of time and computation costs (Desislavov et al., 2021). Therefore, efforts towards decreasing the inference cost in production have increased rapidly over the past few years.

From a system engineering perspective, caching is a standard practice to improve the performance of software systems, helping to avoid redundant computations. Caching is the process of storing recently observed information to be reused when needed in the future, rather than re-computation (Wessels, 2001; Maddah-Ali & Niesen, 2014). Caching is usually orthogonal to the underlying procedure, meaning that it is applied by observing the inputs and outputs of the target procedure and does not engage with the internal computations of the cached function.

Caching effectiveness is best observed when the cached procedure often receives duplicated input while in a similar internal state, for instance, accessing a particular memory block, loading a web page, or fetching the books listed in a specific category in a library database. It is also possible to adopt a standard caching approach with DNNs (e.g., some work caches a DNN's output solely based on its input values (Crankshaw et al., 2017)). However, it would most likely provide a meager improvement due to the high dimension and size of the data (such as images, audios, texts) and low duplication among the requests.

However, due to the feature extraction nature of deep neural networks, we can expect the inputs with similar outputs (e.g., images of the same person or the same object) to have a pattern in the intermediate layers' activation values. Therefore, we exploit the opportunity to cache a DNN's output based on the intermediate layer activation values. In this way, **we can cache the results not by looking at the raw inputs but by looking at their extracted features in the intermediate layers within the model's forward-pass**.

The intermediate layers often have dimensions even higher than the input data. Therefore, we use shallow classifiers (Kaya et al., 2019) to replace the classic cache-storing and look-up procedures. A shallow classifier is a supplementary model attached to an intermediate layer in the base model that uses the intermediate layer's activation values to infer a prediction. In the caching method, training a shallow classifier on a set

of samples mimics the procedure of storing those samples in a cache storage, and inferring for a new sample using the shallow classifier mimics the look-up procedure.

In this paper, we propose early exiting the predictions made by standard classification models using shallow classifiers trained using the samples and information collected at inference time. We first evaluate the rationality of SelfXit in our first research question by measuring how it affects the final accuracy of the given base models and assessing the effectiveness of the parameters we introduce (tolerance and confidence thresholds) as a knob to control the early exit certainty. We further evaluate the method in terms of computational complexity and inference latency improvements in the second and third research questions. We measure these improvements by comparing the FLOPs count, memory consumption, and inference latency for the original model vs. the early exit-enabled version that we built throughout this experiment. We observed a reduction of up to 58% in FLOPs, an acceleration of up to 46% in inference latency while inferring on the CPU and up to 18% on GPU, with less than 2% drop in accuracy. SelfXit demonstrates remarkable performance across a range of models and datasets. By averaging across all confidence levels greater than 0.25, for the simplest model and dataset, CIFAR10-resnet50, it offers a substantial reduction of 52.2% and 32.4% in latency, with only a minor decrease of 6.9% and 1.6% in accuracy compared to the BranchyNet and GATI methods, respectively. Furthermore, in the case of the more complex CIFAR100-Resnet50 model and dataset, our method achieves a significant reduction in latency of 51.6% and 30.4%, while simultaneously enhancing the accuracy by 2.31% and 0.72% compared to the GATI and BranchyNet methods.

In summary, the contributions of this paper are:

- Proposing an early exiting method for the predictions made by off-the-shelf image classifiers and object detection models, which only uses unlabelled samples collected at inference time.

- Automate the process of designing the supplementary models used for early exiting in vision applications and tuning their parameters used to determine early exit hits, using AutoML methods.

- Empirically evaluating the proposed early exiting method using six publicly available off-the-shelf models on six datasets (CIFAR-10, CIFAR-100, ImageNet, LFW, Cityscapes, Criteo), in terms of computational complexity and inference time reduction.

In the remainder of the paper, we discuss the background and related work in Section 2, details of the method in Section 3, the design and evaluation of the study in Section 4, and conclude the discussions in Section 5.

## 2 Related Works

In this section, we briefly review the topics related to the model inference optimization problem. Following this, we introduce the techniques used to build the early exit procedure.

### 2.1 Inference Optimization

There are two perspectives that address the model inference optimization problem. The first perspective focuses on optimizing the model deployment platform and covers a broad range of optimization goals (Yu et al., 2021). These studies often target deployment environments in resource-constrained edge devices (Liu et al., 2021; Zhao et al., 2018) or resourceful cloud-based devices (Li et al., 2020a). Others focus on hardware-specific optimizations (Zhu & Jiang, 2018) and inference job scheduling (Wu et al., 2020).

The second perspective is focused on minimizing the model's inference compute requirements by compressing the model. Among model compression techniques, model pruning (Han et al., 2015; Zhang et al., 2018; Liu et al., 2019b), model quantization (Courbariaux et al., 2015; Rastegari et al., 2016; Nagel et al., 2019), and model distillation (Bucila et al., 2006; Polino et al., 2018; Hinton et al., 2015) are widely used. These ideas alleviate the model's computational complexity by pruning the weights, computing the floating-point calculations at lower precision, and distilling the knowledge from a teacher (more complex) model into a student (less complex) model, respectively. These techniques modify the original model and often cause a

fixed amount of loss in test accuracy. In (Liu et al., 2019a) and (Li et al., 2020b), the authors use search algorithms for channel pruning and suggest compressed networks with lighter models, which lead to lower accuracy but faster inference. In fact, in most cases in these studies, they either achieve low latency but at the cost of accuracy, or reasonable accuracy with high latency. The main disadvantage of these methods is that the computation cannot be tuned with any parameters. In practice, though, to what extent one wants to sacrifice accuracy for faster inference is project dependent and must be tunable.

As we mentioned in the motivation, accelerating inference is crucial for sensitive real-time approaches such as autonomous vehicles. Most of the methods focus on partitioning and offloading calculations to the edge (Mohammed et al., 2020). However, achieving faster decisions for a vehicle to detect a pedestrian requires a more immediate reaction than outsourcing data to the edge. We have applied SelfXit to state-of-the-art object detection approaches, such as Mask R-CNN (He et al., 2017) using popular urban datasets, showing a significant improvement even when using unlabeled data.

## 2.2 Early-Exits in DNNs

"Early exit" generally refers to an alternative path in a DNN model which can be taken by a sample instead of proceeding to the next layers of the model. Many previous works have used the concept of early exit for different purposes (Xiao et al., 2021; Scardapane et al., 2020; Matsubara et al., 2022; Haseena Rahmath et al., 2023). Panda et al. (2016) is one of the early works in this area. They tried to terminate classification by cascading a linear network of output neurons for each convolutional layer and monitoring the output of the linear network to decide about the difficulty of input instances and conditionally activate the deeper layers of the network. But they have not mentioned anything about the inference time and accuracy/time trade-off issue. BranchyNet (Teerapittayanon et al., 2016; Pacheco et al., 2021; Ebrahimi et al., 2022) also utilize their previous observation that features were learned in an early layer of a network to make an early exit. However, they require labeled data to train their models, rendering them unsuitable for use with unlabeled data. Shallow Deep Networks (SDN) (Kaya et al., 2019) points out the "overthinking problem " in deep neural networks. "Overthinking" refers to models spending a fixed amount of computational resources for any query sample, regardless of their complexity (i.e., how deep the neural network should be to infer the correct prediction for the sample). Their research proposes attaching shallow classifiers to the intermediate layers in the model to form the early exits. Each shallow classifier in SDN provides a prediction based on the values of the intermediate layer to which it is attached. (Hanawal et al., 2022) propose an unsupervised early exit method specifically designed for NLP tasks using Elastic BERT. Their approach utilizes a bandit-based learning algorithm (UEE-UCB) to dynamically select exit points in multi-exit DNNs without requiring labeled data. However, the exit points are selected from a fixed set of candidate exits, which may not be optimal in all scenarios. In addition, exit models are kept as simple classifiers, limiting their overall performance. This approach is primarily tailored to BERT-based architectures, making it less applicable to more complex domains, such as vision-based tasks, which are the focus of our work.

On the other hand, Xiao et al. (2021) incorporates the shallow classifiers to obtain multiple predictions for each sample. In their method, they use early exits as an ensemble of models to increase the base model's accuracy.

The functionality of the shallow classifiers in our proposed method is similar to that of SDN. However, the SDN method trains the shallow classifier using ground truth data from the training set and ignores the available knowledge in the original model. This constraint renders the proposed method useless when using a pre-trained model without access to the original training data, which is commonly the case for practitioners.

## 2.3 DNN Distillation and Self-distillation

Among machine learning tasks, the classification category is one of the significant use cases where DNNs have been successful in recent years. Classification is applied to a wide range of data, such as classification of images (Bharadi et al., 2017; Xia et al., 2021), text (Varghese et al., 2020), audio (Lee et al., 2009), and time series (Zheng et al., 2014).

Knowledge distillation(KD) (Bucila et al., 2006; Polino et al., 2018; Hinton et al., 2015) is a model compression method that trains a relatively small (less complex) model known as the student to mimic the behavior of a larger (more complex) model known as the teacher. Classification models usually provide a probability distribution (PD) representing the probability of the input belonging to each class. KD trains the student model to provide similar PDs (i.e., soft labels) to the teacher model rather than training it with just a class label for each sample (i.e., hard labels). KD uses specialized loss functions in the training process, such as Kullback-Leibler Divergence (Lovric, 2011) to measure how one PD is different from another.

KD is usually a 2-step process consisting of training a large complex model to achieve high accuracy and distilling its knowledge into a smaller model. An essential challenge in KD is to choose the right teacher and student models. Self-distillation (Zhang et al., 2021) addresses this challenge by introducing a single-step method to train the teacher model along with multiple shallow classifiers. Each shallow classifier in self-distillation is a candidate student model, which is trained by distilling the knowledge from one or more of the deeper classifiers. In contrast to SDN, self-distillation utilizes knowledge distillation to train shallow classifiers. However, it still trains the base model from scratch along with the shallow classifiers, using the original training set. This training procedure conflicts with our objectives in both aspects. Specifically, we use a pre-trained model and keep it unchanged throughout the experiment and only use inference data to train the shallow classifiers.

In Leontiadis et al. (2021) the authors present a method for enhancing CNN efficiency through early exits. Using supervision, self-supervision, and self-distillation, you can personalize the device, using both labeled and unlabeled data. This allows for dynamic adaptation with varying data availability, focusing on training enhancements. Our work is different by not altering the main model, but instead utilizing early exit layers updated solely during the inference time. This is done to improve latency, without the need for labeled data during these updates, which offers a modular solution with minimal modifications to existing systems.

Our work modifies and puts the presented methods in SDN and self-distillation in the context of early exiting the final predictions of pre-trained DNN models. The method trains the shallow classifiers using only the unlabeled samples collected at run-time and measures the improvement in inference compute costs achieved by the early exits throughout the forward passes.

### 2.4 DNN Prediction Early Exiting

Clipper (Crankshaw et al., 2017) is a serving framework that incorporates early exiting DNNs predictions based on their inputs. Freeze inference (Kumar et al., 2019) investigates the use of traditional ML models such as K-NN and K-Means to predict based on intermediate layer values. They show that the size and computation complexity of those ML models grows proportionally with the number of available samples, and their computational overheads far exceed any improvement. In the Learned Early Exits, Balasubramanian et al. (2021) extend the Freeze Inference by replacing the ML models with a pair of DNN models. A predictor model that predicts the results and a binary classifier that predicts whether the result should be used as the final prediction. Their method uses ground-truth data in the process of training the predictor and selector models. In contrast, our method 1) only uses unlabeled inference data, 2) automates the process of early exit-enabling, 3) uses a confidence-based early exit hit determination, and 4) handles batch processing by batch shrinking.

## 3 Methodology

In this section, we explain the method to convert a pre-trained deep neural model (which we call the backbone) to its extended version with our early exiting method (called early exit-enabled model). The early exiting method adds one or more early-exit paths to the backbone, controlled by the shallow classifiers (which we call the early exit models), allowing the model to infer a decision faster at run-time for some test data samples (early exit hits). Faster decisions for some queries will result in a reduced mean response time.

"Early Exit model" is a supplementary model that we attach to an intermediate layer in the backbone, which given the layer's values provides a prediction (along with a confidence value) for the backbone's output. Just a reminder that as our principal motivation, we assume that the original training data is unavailable for the

user, as is the case for most large-scale pre-trained models used in practice. Therefore, in the rest of the paper, unless we explicitly mention it, the terms dataset, training set, validation set, and test set all refer to the whole available data at run-time or a respective subset.

Our procedure for enabling a pre-trained model to early exit is derived primarily from the self-distillation method (Zhang et al., 2021). However, we adapt this method to early exit-enable pre-trained models using only their recorded outputs through self-training, without access to the ground truth (GT) labels. The novel aspects of our approach consist of selecting the best early exit model through Neural Architecture Search (NAS) followed by an inference-time early exit training. This approach helps us refine our early exit strategy based on performance feedback, with the aim of optimizing both accuracy and inference speed across all stages of the network. Since we employ early exit models in a sequence, if an early early exit model wrongly triggers an incorrect early exit, later early exit models will not have a chance to engage even if they would have been successful. Therefore, an individual early exit model's accuracy/hit rate does not guarantee overall good results since the final outcomes depend on the earlier early exit model's performance as well.

**Setting.** Our approach is built on a pre-trained model, and before starting, we use a NAS algorithm to select the top candidate layers with the potential to cache what the base model has learned. After this selection, the inference process begins by passing an input $i$ through a sequence of layers $l_1, l_2, \ldots, l_M$. Each candidate layer has an associated cache model that performs an early exit, producing an output with the same shape as the base model's output. The base model is frozen during the training and testing phases. For training, in each selected layer $L$, we train the corresponding cache model using the probability distribution $P = B(i)$ generated by the base model as a reference distribution and the probability distribution $Q = B_{\mathrm{L}}(i)$ provided by the cache model for KL divergence. After training these early exit layers, during the testing phase, the model evaluates in each candidate layer whether the current representation $B_{\mathrm{L}}(i)$ is sufficient to make a confident prediction and potentially terminate the inference process early. This decision is based on the confidence measure calculated in each layer. The process continues through the layers until an early exit is triggered or the final layer $l_M$ is reached, where the final prediction is made. The model determines the output by $\mathrm{argmax} B_{\mathrm{L}}(i)$.

A step-by-step guide on early exit-enabling an off-the-shelf pre-trained model from a user perspective contains the following steps:

1. **Identify the candidate layers to be early exit** This step involves analyzing each selected model to identify potential positions for early exit layers. Layers whose outputs are independent of other layers' states are chosen as candidates, enabling training possibilities.

2. **Build an early exit model for each candidate** Using neural architecture search (NAS), we evaluate all possible subsets for these early exit layers. NAS also scores the different architectures defined for our early exit models.

3. **Assign confidence thresholds to built models to determine early exit hits** Each early exit model features a softmax as the final layer. In this step, we assign confidence thresholds to the constructed models to determine early exit hits. The probability value associated with the predicted output reflects the model's confidence in its prediction. This confidence level for a given input determines whether we accept that prediction as an early exit hit or continue processing through the rest of the backbone model.

4. **Evaluate and optimize the early exit-enabled model** In this step, we evaluate and optimize the early exit-enabled model in the inference mode and also present our algorithm to update all early exit layers.

5. **Early-Exit Optimization Implementation** In this step, we implement the algorithm and train all early exit models using self-training (without using any ground truth) in the datasets we have mentioned.

In subsections 3.1 to 3.5, we further discuss the procedure and design decisions in each step outlined above.

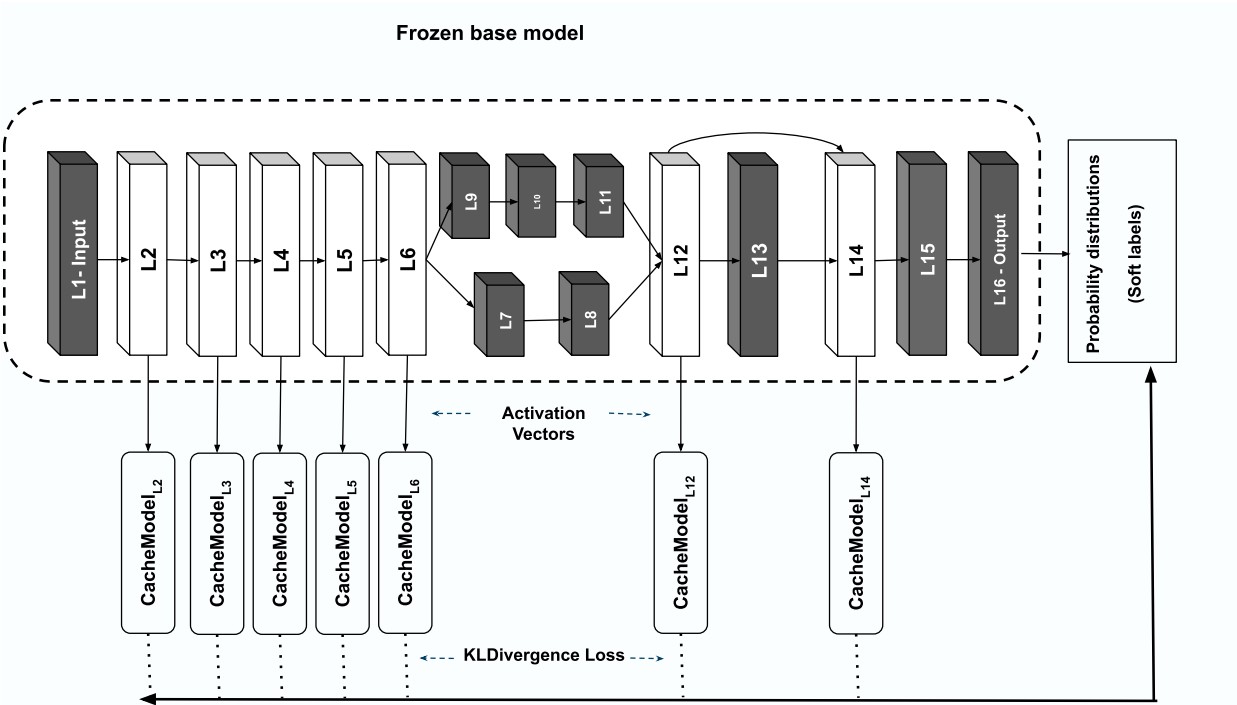

Figure 1: Early Exit-enabling procedure, candidate layers, and data paths.

## 3.1 Identifying candidate layers

Choosing which layers to early exit is the first step towards early exit-enabling a model. A candidate layer is a layer that we will examine its values' correlation to the final predictions by training an early exit model based on them. One can simply list all the layers in the backbone as candidates. However, since we launch a search for an early exit model per candidate layer in the next step, we suggest narrowing the list by filtering out some layers with the following criteria:

- Some layers are disabled at inference time, such as dropouts and batch normalizations. These layers do not modify their input values at inference time. Therefore, we cross them off the candidate list.

- Some of the last layers in the model (close to the output layer, such as L15 in Figure 1) might not be valuable candidates for early exiting, since the remaining layers might not have heavy computations to reach the output. This is particularly important as each early exit model typically consists of at least two layers. If the main path has only two layers remaining before the final result and softmax, adding an early exit at that point would not significantly improve inference time. Therefore, we discard the last two layers (e.g., L15 and L16) from consideration.

- DNN models usually are composed of multiple components (i.e. first-level modules) consisting of multiple layers such as multiple residual blocks in ResNet models He et al. (2016)). We narrow down the search space to the outputs layers in those components.

- We only consider the layers, for which, given their activation values, the backbone's output is uniquely determined without any other layer's state involved (i.e., the backbone's output is a function of the layer's output). In other words, a layer with other layers or connections in parallel (such as L7–L11 and L13 in the Figure 1) is not suitable for early exiting, since the backbone's output does not solely depend on the layer's output. While this approach has proven effective for the architectures we tested, we acknowledge that it may not generalize well to more complex architectures like Mixture of Experts (MoE) models, where parallel experts are common. We have added this as a limitation of our method and suggest future exploration in this area.

Having the initial set of the candidate layers, we next build and associate an early exit model to each one.

## 3.2 Building early exit models

Building an early exit model to be associated with an intermediate layer in the backbone consists of finding a suitable architecture for the early exit model and training the model with that architecture. The details of the architecture search (search space, search method, and evaluation method) and the training procedure (training data extraction and loss function) are discussed in the following two subsections.

### 3.2.1 Early Exit models architecture

An early exit model can have an architecture of any depth and breadth size, as long as it provides more computational improvement than its overhead. In other words, it must have substantially less complexity (i.e., number of parameters and connections) than the rest of the layers in the backbone that come after the corresponding intermediate layer. The search space for such models would contain architectures with different numbers and types of layers (e.g., a stack of dense and/or convolution layers). However, all models in the search space must produce a PD identical to the output of the backbone in terms of size (i.e, the number of classes) and activation (e.g., SoftMax or LogSoftMax).

In our experiments, the search space consists of architectures with a stack of (up to 2) convolution layers followed by another stack of (up to 2) linear layers, with multiple choices of kernel and stride sizes for the convolutions and neuron counts for the linear layers. However, users can modify or expand the search space according to their specific needs and budget.

The objective of the search is to find a minimal architecture that converges and predicts the backbone's output with acceptable accuracy. Note that any accuracy given by an early exit model (better than random) can be helpful, as we will have a proper selection mechanism later in the process to only use the early exit predictions that are (most likely) correct, and also to discard the early exit models yielding low computational improvement.

The user can conduct the search by empirically sampling through the search space or by using a automated Neural Architecture Search (NAS) tool such as Auto-Keras (Jin et al., 2019), Auto-PyTorch (Zimmer et al., 2021), Neural Network Intelligence (NNI) (Microsoft, 2021), or NASLib (Ruchte et al., 2020). However, we used NNI to conduct the search and customized the evaluation process to account for the models' accuracy and their computational complexity. We have used the floating point operations (FLOPs) count as the estimation for the models' computational complexity in this stage.

Several factors influence the architecture of an early exit model for a given intermediate layer. These factors include the dimensions of the intermediate target layer, its position in the backbone, and the data set specifications, such as its number of target classes. For example, the first early exit models in the CIFAR100-Resnet50 and CIFAR10-Resnet18 experiments (shown as Cache 1 in Figure 6) have the same input size, but since CIFAR100 has more target classes, it reasonably requires an early exit model with more learning capacity. The architecture of each early exit model is further adjusted based on the specific requirements of the dataset, particularly the number of output classes. For example, while the early exit models for CIFAR10 and CIFAR100 share similar input sizes, the early exit models for CIFAR100 are designed with increased capacity to handle its larger number of classes. This adaptation is guided by the suggestions provided by the Neural Architecture Search (NAS), ensuring that each early exit model is optimized for the dataset to which it is applied. This approach is consistent across different datasets, such as ImageNet, where the early exit models are similarly adjusted to accommodate the 1,000 output classes. These architectural adjustments ensure that the models remain efficient and capable of handling the varying complexities of different datasets. Therefore, using NAS to design the early exit models helps automate the process and alleviate deep learning expert supervision in designing the early exit models.

NAS tries to minimize the total accuracy no more than the tolerance, so for all possible subsets, there will be a maximum range. This method does not guarantee the best score, but is a good solution. Subsets with the best score will be selected as our early exit system which will be added inside the model to be trained individually by the predictions and perform early exits.

Regardless of the search method, evaluating a nominated architecture requires training a model with the given architecture, of which we discuss the procedure in the next section. Moreover, since the search space is limited in depth, it is possible that, for some intermediate layers, neither of the early exit models converges (i.e., the model provides nearly random results). In such cases, we discard the candidate layer as non-suitable for early exiting.

### 3.2.2 Training an early exit model

Figure (1) illustrates the schema of an early exit-enabled model consisting of the backbone (the dashed box) and the associated early exit models. The objective of an early exit model is to predict the output of the backbone model, given the corresponding intermediate layer's output, per input sample.

Similarly to the backbone, early exit models are classification models. However, their inputs are the activation values in the intermediate layers. As suggested in self-distillation (Zhang et al., 2021), training an early exit model is essentially similar to distilling the knowledge from the backbone (final classifier) into the early exit model.

Therefore, to distill the knowledge from the backbone into the early exit models, we need a medial data set (MD) based on the collected inference data (ID). The medial data set for training an early exit model associated with an intermediate layer L in the backbone B consists of the set of activation values in the layer L and the PDs given by B per samples in the given ID, formally annotated as follows:

$$MD_L = [i \in ID :< B_L(i), B(i) >] \tag{1}$$

where:

$MD_L$ : Medial dataset for the early exit model associated with the layer L
ID     : The collected inference data consisting of unlabeled samples
$B_L(i)$: Activation values in layer L given the sample i to the backbone B
B(i)   : The backbone's PD output for the sample i

Note that the labels in MDs are the backbone's outputs and not the GT labels, as we assumed the GT labels to be unavailable. We divide $MD_L$ into three partitions ($MD_L^{Train}, MD_L^{Val}, MD_L^{Test}$) and use them, respectively, similarly to the common deep learning training and test practices.

Similarly to the distillation method (Hinton et al., 2015), we use the Kullback-Leibler divergence (KLDiv) (Lovric, 2011) loss function in the training procedure. KLDiv measures how different the two given PDs are. Thus, minimizing the value of the KLDiv loss in $MD_L^{Train}$ trains the early exit model to estimate the prediction of the backbone ($B(i)$).

Unlike self-distillation, where (Zhang et al., 2021) train the backbone and shallow classifiers simultaneously, in our method, while training an early exit model, it is crucial to freeze the rest of the model including the backbone and the other early exit models (if any) in the collection, to ensure that the training process does not modify any parameter not belonging to the current early exit model.

### 3.3 Assigning confidence threshold

The probability value associated with the predicted class (the one with the highest probability) is known as the confidence of the model in the prediction. The early exit model's prediction confidence for a particular input will indicate whether we stick with that prediction (early exit hit) or proceed with the rest of the backbone to the next — or probably final — exit (early exit miss).

Confidence calibration means improving the model to provide accurate confidence. In other words, the confidence of a well-calibrated model accurately represents the likelihood for that prediction to be correct(Guo et al. (2017)). An overconfident early exit model will lead the model to prematurely exit for some samples based on incorrect predictions, whereas an underconfident early exit model will bear a low early exit hit rate. Therefore, after building an early exit model, we also calibrate its confidence using $MD_L^{Val}$ to better

Table 1: Early Exit prediction confusion matrix, C: Early Exit predicted class, B: Backbone's predicted class, GT: Ground Truth label

| Category | B = C | B = GT | C = GT |
|:---:|:---:|:---:|:---:|
| $BC$ | ✓ | ✓ | ✓ |
| $\overline{B}C$ | ✓ | X | X |
| $B\overline{C}$ | X | ✓ | X |
| $\overline{B}C$ | X | X | ✓ |
| $\overline{B}\ \overline{C}$ | X | X | X |

distinguish the predictions that are more likely to be correct. Several confidence calibration methods are discussed in Guo et al. (2017), among which temperature scaling (in the output layer) has been shown to be practical and easy to implement.

Having the model calibrated, we next assign a confidence threshold value to the model which will be used at inference time to determine the early exit hits and misses. When an early exit model identifies an early exit hit, its prediction is considered the final prediction. However, when needed for validation and test purposes, we obtain the predictions from the early exit model and the backbone.

Confidence calibration involves adjusting the predictive confidence of our early exit models to more accurately reflect their true performance, particularly how likely they are to be correct. This is crucial for making reliable decisions during inference, especially when early exits from the model are considered based on these confidence scores. To calibrate the confidence levels of our early exit models, we employ a threshold that measures confidence based on the model output during the validation phase. Specifically, each early exit model functions as a classifier: during both the validation and testing phases, it generates an output that carries an associated confidence value. This confidence value indicates the probability that the model prediction is correct.

An early exit model's prediction (C) for an input to the backbone falls into one of the 5 correctness categories listed in table 1 with respect to the ground truth labels (GT) and the backbone's prediction (B) for the input.

Among the cases where the early exit model and the backbone disagree, the $B\overline{C}$ predictions negatively affect the final accuracy, and on the other hand, the $\overline{B}C$ predictions positively affect the final accuracy. Equation 2 formulates an actual effect of an early exit model on the final accuracy.

$$F_\Delta(\theta) = \overline{B}C_\Delta(\theta) - B\overline{C}_\Delta(\theta) \tag{2}$$

Where:

$\Delta$ : The early exit model
$F_\Delta$ : The actual accuracy effect $\Delta$ causes given $\theta$ as threshold
$B\overline{C}_\Delta$ : Ratio of $B\overline{C}$ predictions by $\Delta$ given $\theta$ as threshold
$\overline{B}C_\Delta$ : Ratio of $\overline{B}C$ predictions by $\Delta$ given $\theta$ as threshold

However, since we used the unlabeled inference data to form the MDs, we can only estimate an upper bound for the effect of the early exit model in the final accuracy. The estimation assumes that an incorrect early exit would always lead to an incorrect classification of the sample ($\overline{B}C$). We estimate the change in the accuracy upper bound an early exit model causes given a certain confidence threshold by its hit rate and early exit accuracy:

$$F_\Delta(\theta) \leq HR_\Delta(\theta) \times (1 - CA_\Delta(\theta)) \tag{3}$$

Where

$\Delta$ : The early exit model

$F_\Delta$ : The expected accuracy drop $\Delta$ causes given $\theta$ as threshold

$HR_\Delta$ : Hit rate provided by $\Delta$ given $\theta$ as threshold

$CA_\Delta$ : Early Exit accuracy provided by $\Delta$ given $\theta$ as threshold

Given the tolerance $T$ for the drop in final accuracy, we assign a confidence threshold to each early exit model that yields no more than the expected drop in accuracy $T/2^n\%$ in $MD_L^{Val}$ according to equation 3, where n is the 1-based index of the early exit model in the setup.

It is important to note that there are alternative methods to distribute the accuracy drop budget among the early exit models. For example, one can equally distribute the budget. However, as we show in the evaluations later in Section 4.6.1, we find it reasonable to assign more budget to the early exit models shallower positions in the backbone.

### 3.4 Evaluation and optimization of the early exit-enabled model

So far, we have a set of early exit layers and their corresponding early exit models ready for deployment. The algorithm 1 demonstrates a pseudoimplementation of a Python-style inference process for the model enabled for early exit. When the early exit-enabled model receives a batch of samples, it proceeds layer-by-layer similarly to the standard forward-pass. Once an early exit layer's activation values are available, it will pass the values to the corresponding early exit model and obtain an early prediction with a confidence value per sample in the batch. For each sample, if the corresponding confidence value exceeds the specified threshold, we consider it an early exit hit. Hence, we have the final prediction for the sample without passing it through the rest of the backbone. At this point, the prediction can be sent to the procedure awaiting the results (e.g., an API, a socket connection, or a callback). We shrink the batch by discarding the early exit hit items at each exit and proceed with a smaller batch to the next (or the final) exit.

---

**Algorithm 1** Early Exit-enabled model inference

---

**Require:** Backbone                                             ▷ The original model

**Require:** Early Exit Layers                             ▷ List of early exit layers

**Require:** Layer          ▷ As part of Backbone, including the associated early exit model and threshold

  1: **procedure** FORWARDPASS(X, callback)                      ▷ X: Input batch

  2:     **for** Layer in Backbone.Layers **do**               ▷ In order of presence[1]

  3:         X ← Layer(X)

  4:         **if** Layer in Early Exit Layers **then**

  5:             Early Exit ← Layer.Early ExitModel

  6:             T ← Early Exit.Threshold

  7:             early exit PDs ← Early Exit(X)

  8:             confidences ← max(early exit PDs, axis=1)

  9:             callback(early exit PDs[confidences≥ T])         ▷ Resolve early exit hits

10:             X ← X[confidences<T]                      ▷ Shrink the batch

11:         **end if**

12:     **end for**

13: **end procedure**

---

So far in the method, we have only evaluated the early exit models individually, but to gain the highest improvement, we must also evaluate their collaborative performance within the early exit-enabled model. Once the early exit-enabled model is deployed, each early exit model affects the hit rates of the following early exit models by narrowing the set of samples for which they will infer. More specifically, even if an early exit model shows promising hit rate and accuracy in individual evaluation, its performance in the deployment can

---

[1]The loop is to show that each early exit model will receive the early exit layer's activation values when available, immediately, before proceeding to the next layer in the base model.

be affected due to the previous early exit hits made by the earlier early exit models (connected to shallower layers in the backbone). Therefore, we need to choose the optimum subset of early exit models to infer the predictions with the minimum computations.

A brute-force approach to find the optimum subset would require evaluating the early exit-enabled model with each subset of the early exit models. However, we implement a more efficient method without multiple executions of the early exit-enabled model.

First, for each early exit model, we record its prediction per sample in the $MD_L^{Val}$ and their confidence values. We also record two FLOPs counts per early exit model; one is the early exit model's FLOPs count($C_1$), and the other is the fallback FLOPs count, which denotes the FLOPs in the remaining layers in the backbone($C_2$). For example, for the layer `L12` in the Figure 1, $C_1$ is the FLOP count of the corresponding early exit model and $C_2$ is the FLOP count in the layers `L13` through `L16`.

For each subset $S$, we process the lists of predictions recorded for each model in $S$ to generate the lists of samples that they actually receive when deployed along with other early exit models in $S$. The processing consists of keeping only the samples in each list for which there have been no early exit hits by the previous early exit models in the subset. Further, we divide each list into two parts according to each early exit model's confidence threshold; one consisting of the early exit hits, and the other consisting of the early exit misses.

Finally, we score each subset using the processed lists and recorded values for each early exit model in $S$ as follows:

$$K(S) = \sum_{\Delta \in S} |H_\Delta| \times (C_{2,\Delta} - C_{1,\Delta}) - |M_\Delta| \times C_{1,\Delta} \qquad (4)$$

Where

$K$ : The early exiting score for subset $S$
$\Delta$ : An early exit model in $S$
$H_\Delta$ : The generated list of early exit hits for $\Delta$
$M_\Delta$ : The generated list of early exit misses for $\Delta$
$C_{1,\Delta}$ : FLOPs count recorded for $\Delta$
$C_{2,\Delta}$ : Fallback FLOPs count recorded for $\Delta$

The score equation accounts for both the improvement an early exit model provides through its early exit hits within the subset and the overhead it produces for its early exit misses.

Using additional early exit layers may cause earlier detection of a class with a higher probability (with the use of an appropriate confidence for early exit layers) before moving to the next layers, so the total accuracy would increase with the cost of memory consumption.

The final schemas after applying the method on MobileFaceNet, EfficientNet, ResNet18, and ResNet50 are discussed in the main text, with detailed illustrations provided in the Appendix (see Figure 6). This figure demonstrates the chosen subsets and their associated early exit models for each backbone and data set.

### 3.5 Early-Exit Optimization Implementation

In this section, we present the implementation and application of early exit models for efficient and timely predictions in three distinct tasks: image classification, object detection, and recommendation systems. We begin by incorporating our proposed early exit approach into architectures widely used for image classification tasks, i.e. MobileFaceNet, EfficientNet, ResNet18, and ResNet50, on benchmark datasets, i.e. CIFAR10, CIFAR100, ImageNet and LFW.

Inspired by the promising results obtained in image classification, we further extend our methodology to address a critical real-world scenario: pedestrian detection in urban environments. For this purpose, we adopt the state-of-the-art Mask R-CNN model, renowned for its exceptional object detection capabilities.

By integrating our early exit strategy into Mask R-CNN, we enable the model to detect pedestrians at an earlier stage during inference, thus significantly reducing the processing time and providing timely warnings to autonomous vehicles about the presence of pedestrians within a given scene.

The significance of our contributions lies in the potential to enhance the safety and responsiveness of autonomous vehicles in urban settings, where pedestrian detection plays a pivotal role in avoiding accidents and ensuring seamless interaction between vehicles and pedestrians. In our approach, while prioritizing accuracy, we forgo an important aspect: the exact coordinates of the detected objects. By implementing early exit models triggered by the detection of pedestrians or humans, our goal is to achieve faster processing and response times. However, we acknowledge that, in certain cases, reacting within the required time window is of utmost importance. The trade-off between accuracy and response time is a crucial consideration in our methodology, and we recognize the significance of timely actions, especially in scenarios where immediate responses are critical to ensure optimal outcomes.

In the context of the Mask R-CNN model, various options are available to select different backbones and settings, allowing flexibility in performance evaluation and adaptation to specific tasks. While numerous configurations are possible, we opted to utilize a publicly available, pre-trained backbone to ensure that our experiments are standardized and well established. This choice allows us to focus on the effectiveness of our proposed approach, taking advantage of the robustness and generalization capabilities of the chosen backbone. Additionally, using a pre-trained model helps to mitigate potential biases in training data and enables fair comparisons with other methods that have adopted similar backbones. The final schema of the early exit models for Mask R-CNN with Resnet50 backbone is illustrated in the Appendix (see Figure 7). We extend our early exit classification model to implement pedestrian object detection. The object detection model scans an input image and detects multiple objects within the image, assigning each detected object a probability distribution over possible classes.

In object detection, the model's task is to identify various objects within an image and to provide a probability distribution for each detected object. The challenge therein lies in determining an effective method for updating the convolutional dense layer early exits in this context. Finally, we explore the application of our early exit strategy in recommendation systems. Specifically, we integrate early exit nodes into the Deep Learning Recommendation Model (DLRM), as suggested by the Microsoft Neural Architecture Search (NAS) algorithm. This integration aims to optimize the inference time while maintaining high recommendation accuracy. The schema of the DLRM architecture with early exits is detailed in Figure 8.

### 3.5.1 Updating Early Exits for Pedestrian Detection

In object detection, pedestrians are one of the most common classes. To optimize the performance of our early exit framework for pedestrian detection, we explore three update strategies for the layers.

- Updating with the most confident pedestrians: In this approach, we selectively update the layers with features extracted from regions that are confidently classified as pedestrians. By focusing on the most confident detections, we aim to enhance the early exit memory's relevance to crucial features associated with pedestrians in the scene.

- Updating with the most confident object: We investigate updating the layers with features from regions classified as the most confident object, regardless of whether it is a person or another class. This strategy is designed to ensure that the early exit memory reflects critical features representative of the dominant object class in the scene.

- Updating with all detected objects: In this method, we update the layers with features from all detected objects in the scene. While this approach may provide a broader context, it may introduce redundancy and bias towards the more prevalent classes.

After testing these three early exit-updating approaches, the results supported updating with the most confident single object as the best-performing method. Training early exit layers with a sole focus on individual objects, such as pedestrians, leads to non-convergence and a lack of meaningful learning. Even

prior to testing, it became evident that training layers exclusively with a defined class introduce bias, impeding effective learning. Meanwhile, training early exit layers using a diverse set of objects results in model confusion, manifesting itself as reduced accuracy in our testing outcomes. So, our selected approach updates the model with the most certain detection while avoiding the issues of bias and multi-object confusion.

## 4 Empirical Evaluation

In this section, we explain the objective of our experiment, the research questions, the implementation of the tool and the design of the experiment including the backbones and data sets, the evaluation metrics, and the configuration of the environment.

### 4.1 Objectives and research questions

The high-level objective of this experiment is to assess the ability of the automated layer early exiting mechanism to improve the computation requirements and inference time for DNN-based services.

To address the above objective, we designed the following research questions (RQ):

RQ1 To what extent can early exit models accurately predict the backbone's output and the ground truth data? This RQ investigates the core idea of early exiting as a mechanism to estimate the final output earlier in the model. The assessments in this RQ consider the early exit models' accuracy in predicting the backbone's output (early exit accuracy) and predicting the correct labels (GT accuracy).

RQ2 To what extent can early exit-enabling improve computing requirements? In this RQ, we are interested in how early exit-enabling affects the models' computation requirements. In these measurements, we measure the FLOPs counts and memory usage as metrics for the models' compute consumption.

RQ3 How much acceleration does early exit enable on the CPU / GPU? In this RQ, we are interested in the actual amount of end-to-end speedup that an early exit enabled model can achieve. We break this result down to CPU and GPU accelerations, since they address different types of computation during the inference phase and thus may have been differently affected.

RQ4 How does the early exit-enabled model's accuracy/latency trade-off compare with other early exit methods? In this research question, our aim is to assess and compare the performance of your early exit-enabled model against other existing early exit methods regarding the trade-off between accuracy and latency in practical, real-world scenarios.

### 4.2 Tasks and datasets

Among the diverse set of real-world classification tasks that are implemented by solutions using DNN models, we have selected two representatives: face recognition and object classification. Both tasks are quite commonly addressed by DNNs and often used in large-scale services that have non-functional requirements such as: high throughput (due to the nature of the service and the large volume of input data) and are time-sensitive.

Face recognition models are originally trained on larger datasets such as MS-Celeb-1M (Guo et al., 2016) and are usually tested with different — and smaller — datasets such as LFW (Huang et al., 2008), CPLFW (Zheng et al., 2017), RFW (Wang et al., 2019), AgeDB30 (Moschoglou et al., 2017), and MegaFace (Kemelmacher-Shlizerman et al., 2016) to test the models against specific challenges, such as age / ethnic biases and recognizing mask-covered faces.

We used the labeled face in the wild (LFW) dataset for face recognition, which contains 13,233 images of 5,749 individuals. We used images from 127 identities that have at least 11 images in the set so we can split them for training, validation, and testing.

We also used CIFAR10, CIFAR100, and ImageNet test sets (Krizhevsky & Hinton, 2009; Russakovsky et al., 2015) for object classification, each containing 10,000 images distributed equally among 10 and 100 classes for CIFARs and 100,000 images among 1,000 classes for ImageNet. CIFARs have a size of $32 \times 32$ pixels, while the ImageNet data set originally contains images of different sizes. To standardize the input size for ImageNet, we used a common method of resizing the images to $256 \times 256$ pixels and then cropping them to $224 \times 224$ pixels.

Reminder that we do not use the training data, rather we only use the test sets to simulate incoming queries at run-time. Specifically, we used only the test splits of the CIFARs and ImageNet datasets. However, we used the entire LFW data as it has not been used to train the face recognition models. Moreover, we do not use the labels in these test sets in the training and optimization process, rather we only use them in the evaluation step to provide GT accuracy statistics.

Each data set mentioned above represents an inference workload for the models. Thus, we split each one into training, validation and test partitions with 50%, 20%, and 30% proportionality, respectively. However, we augmented the test sets using flips and rotations to improve the statistical significance of our testing measurements.

We used the CityScape dataset to assess the presence of pedestrians ((Cordts et al., 2016)). This data set is valuable for our research on the comprehension of urban scenes, as it offers meticulously annotated images, with pixel-level labels, depicting various urban environments from the perspective of a vehicle. To evaluate the accuracy of our model, we needed labeled data for testing purposes. Upon observing that the test dataset within the CityScape dataset contained dummy labels, we opted to utilize the validation subset instead.

To demonstrate the applicability of our early exit model on other nonimage-based tasks, we integrated it into a recommendation system. The data set we use is from the Criteo data set on Kaggle (Jean-Baptiste Tien, 2014), used for the Display Advertising Challenge, which is a comprehensive resource designed to benchmark click-through rate (CTR) prediction algorithms. It includes data collected over a seven-day period, consisting of feature values and click feedback for millions of display ads. The dataset is divided into training and testing parts, with each part containing both clicked and nonclicked examples that have been sub-sampled to manage dataset size. The data set consists of 39 features per sample: 13 integer features, primarily count data, and 26 hashed categorical features for anonymization. The details of these features are not disclosed and may include missing values. This application can show the ability of the model to significantly improve inference speed while managing large-scale data structures.

### 4.3 Backbones and Models

The proposed early exit enabler method is applicable to any deep classifier model. However, the results will vary for different models depending on their complexity.

Among the available face recognition models, we have chosen the well-known MobileFaceNet and EfficientNet models to evaluate the method, and we experiment with ResNet18 and ResNet50 for object classification.

The object classification models are typical classifier models out-of-the-box. However, face recognition models are feature extractors that provide embedding vectors for each image based on the face / location features. They can still be used to classify a face-identity dataset. Therefore, we attached a classifier block to these models and trained them (with the feature extractor layers frozen) to classify the images of the 127 identities with the highest number of images in the LFW dataset (above 10). It is important to note that since the added classifier block is a part of the pre-trained model under study, we discarded the data portion used to train the classifier block to ensure we still hold on to the constraint of working with pre-trained models without access to the original training dataset.

As stated previously, our pedestrian detection approach required the selection of an object detection technique capable of identifying pedestrians within images. We adopted the Mask R-CNN framework. This method encompasses a backbone component (for which we employed ResNet50) and two additional sections that consume significant time and memory resources.

However, for our specific use case of providing early warnings to autonomous vehicles regarding the presence of pedestrians, the precise localization of pedestrians is not essential. Consequently, we chose to disregard the other resource-intensive sections, resulting in substantial time savings while still achieving the necessary level of awareness of pedestrian presence.

The Deep Learning Recommendation Model (DLRM) (Naumov et al., 2019) is a neural network architecture designed for use in recommendation systems. It efficiently handles categorical and numerical data, making it highly effective for personalized recommendation tasks. DLRM uses a combination of embedding tables for categorical features and multi-layer perceptrons (MLPs) for numerical features. These components are then interacted with through a specialized dot product interaction operation, which enables the model to learn and predict complex patterns from user-item interactions.

### 4.4   Metrics and measurements

Our evaluation metrics for RQ1 are ground truth (GT) accuracy and early exit accuracy. Early exit accuracy measures how accurately an early exit model predicts the backbone's output (regardless of correctness). The GT accuracy applies to both the early exit-enabled model and each individual early exit model. However, early exit accuracy only applies to early exit models. Individual early-exit layers' accuracies might be different from the final early-exit-enabled model's accuracy. This discrepancy arises because layer-wise accuracies are calculated only for the cases where the input is classified at that specific layer (hit cases), which typically results in higher accuracies. For example, of 5000 inputs, a particular layer may serve as an early exit for only 5 inputs at a confidence level of 99%. Although this layer may exhibit high accuracy in these cases, it would have only a slight impact on the overall final accuracy of the model.

In RQ2, we compare the original models and their early exit-enabled version in terms of the average FLOP count occurring for inference and their memory usage. We only measure the resources used in the inference. Specifically, we exclude the training-specific layers (e.g. batch normalization and dropout) and computations (e.g. gradient operations) in the analysis.

The FLOP count takes into account the model architecture and input size and estimates the computations required by the model to infer the input (Desislavov et al., 2021). In other words, the fewer FLOPs used for inference, the more efficient the model is in terms of compute and energy consumption.

On the other hand, we report two aspects of memory usage for the models. The first is the total space used to load the models in the memory (i.e, model size). This metric is essentially agnostic to the performance of early exit models and only considers the memory cost of loading them along with the backbone.

In addition to the memory required for their weights, DNNs also allocate a sizeable amount of temporary memory for buffers (also referred to as tensors) that correspond to intermediate results produced during the evaluation of the DNN's layers (Levental, 2022). Therefore, our second metric is the live tensor memory allocations (LTMAs) during inference. LTMA measures the total memory allocated to load, move, and transform the input tensor through the model's layers to form the output tensor while executing the model.

In RQ3, we compare the average inference latency of the original model with its early exit-enabled counterpart. Inference latency measures the time spent from passing the input to the model till it exits the model (by either an early-exit or the final classifier in the backbone). Various factors affect inference latency including hardware-specific optimizations (e.g., asynchronous computation), framework, and model implementation. In our measurements, the framework and model implementations are fixed as discussed in the Appendix A.1. However, to account for other factors, we repeat each measurement 100 times and report the average inference latency recorded for each experiment. Further, to also account for the effects of asynchronous computations in GPU inference latency, we repeated experiments with different batch sizes.

Please refer to Appendix A for details of implementation and setup.

RQ4 aims to evaluate the trade-offs between accuracy, latency, and computational cost of an early exit-enabled model compared to other early exit methods. This inquiry is crucial to understanding the efficiency and effectiveness of different approaches.

To address this question, we have selected two prominent and easy-to-implement early exit methods for comparison. These methods are evaluated using classification tasks in ResNet18, ResNet50, and the CIFAR datasets.

## 4.5 Baselines

In this subsection, we discuss the baselines used to evaluate the proposed unsupervised early exit method. Specifically, we consider two prominent methods: BranchyNet (Teerapittayanon et al., 2016) and GATI (Balasubramanian et al., 2021), both of which aim to reduce the latency of deep neural network (DNN) inference while maintaining high accuracy. These baselines typically require access to labeled training data, which contrasts with our unsupervised approach.

BranchyNet is designed to improve the efficiency of DNN inference by allowing certain samples to exit the network early through additional side branch classifiers. These branches are strategically placed at various depths in the network to enable predictions at different levels of abstraction. During the training phase, BranchyNet jointly optimizes the loss functions of both the main branch and the side branches, effectively regularizing the network and mitigating vanishing gradients. The inference phase utilizes entropy-based thresholds to determine whether a sample should exit early or continue through the deeper layers of the network. In our implementation of BranchyNet as a baseline, we maintained the original supervised setup, but we adapted the training process to ensure a fair comparison. Specifically, we used the same data for training the early exit layers across all methods, which was a portion of the test data set (the other half was not used for evaluation). Furthermore, we froze the base model during the training of the side branches, a crucial difference from the original BranchyNet, which does not freeze the base model. For evaluation, we applied BranchyNet to the remaining half of the test data set reserved for evaluation, allowing us to directly compare its performance against our unsupervised method.

GATI introduces a learned caching mechanism to accelerate DNN inference by exploiting temporal locality in prediction-serving workloads. GATI caches the hidden layer output of the DNN, enabling faster predictions for inputs that exhibit characteristics similar to previously processed data. The system employs simple machine learning models as learned caches, which are continuously updated based on incoming data. Unlike BranchyNet, GATI does not modify the DNN architecture but instead optimizes the inference process by dynamically skipping layers when a cache hit is detected. In our evaluation of GATI, we used a pre-trained base model and trained the cache layers using the same portion of the test dataset as other methods, ensuring consistent data usage. Similarly to BranchyNet, we froze the base model during the training of the caches, differing from GATI's original approach, which does not freeze the base model. While both baselines rely on supervised training, our method is designed to function in an unsupervised manner, particularly in scenarios where a base pre-trained model is accessible, but labeled data may not be available. This flexibility allows our approach to be deployed in contexts where labeled data is scarce or unavailable, leveraging the test set itself for training early exits. To ensure a fair comparison with our method, we adapted both baselines to this framework by using the same data for training and freezing the base model during the training of early-exit layers, which aligns with our approach. However, it is important to note that these baselines do not inherently support unsupervised training, which is a key contribution of our work. An important point to note is that our Base model represents the upper bound of our work, as it is built on a pre-trained model. Our unsupervised method aims to make early-exit results resemble the Base model's predictions rather than converging to the ground truth labels. Consequently, our upper bound is consistently the Base model, while other methods may exhibit varying test accuracies due to additional training and reliance on labeled data. However, this does not necessarily apply to other baselines, such as BranchyNet and GATI, as they may use different approaches for training. Although our Base model demonstrates the best possible performance within the constraints of our approach, the same cannot be directly inferred for the other baselines, which may not achieve the same upper bound due to their reliance on labeled training data and other training techniques.

This is our summarized evaluation process:

- **Pretraining Models:** We use pretrained models, typically trained with both training and validation data. After pretraining, we integrate our layers (early exits) into the models. Initially, these new blocks and layers are not trained.

- **Test Data Splitting:** For testing our method, we split the test data set into two separate parts. We then freeze the main model and update the early exit layers by inferring on the first portion of the test data. During this phase, the main model weights are not updated, only the new early exit layers are trained. Importantly, we do not use the labels from the test data for training or optimization, thus avoiding overfitting concerns.

- **Performance Evaluation:** After training the early exit layers, we test the performance of the entire model using the remaining part of the test dataset. This approach may result in a slight loss of accuracy, but it significantly reduces the total inference time due to the early exits.

### 4.6 Experiment results

In this sub-section, we evaluate the results of applying the method on the baseline backbones and discuss the answers to the RQs. For a more comprehensive understanding, further details about our implementation, including access to the code repository, are provided in the Appendix A.

#### 4.6.1 RQ1. How accurate are early exit models in predicting the backbone output and the ground truth labels?

In this RQ, we are interested in the performance of the built early exit models in terms of their hit rate, GT accuracy, and early exit accuracy. We break down the measurements into two parts. The first part covers the individual performance of the early exit models over the whole test set without any other early exit model involved. The second part covers their collaborative performance within the early exit-enabled model.

#### 4.6.2 Early Exit models' individual performance

Figure 2 shows the individual performance of each early exit model against any confidence threshold value in the CIFAR100-Resnet50 experiment. Figures demonstrating the same measurements for other experiments are available in the Appendix C.

We make three key observations here. First, deeper early exit models are more confident and accurate in their predictions. For example, early exit 1 in Figure 2 has 33.36% GT accuracy and 35.74% early exit accuracy, while these metrics increase to 78.60% and 95.38% for early exit 3, respectively. This observation agrees with the generally acknowledged feature extraction pattern in the DNNs – deeper layers convey more detailed information.

The second key observation is the inverse correlation between the early-exit models' accuracy (both GT and early-exit) and their hit rates. This observation highlights the reliability of confidence thresholds in distinguishing predictions that are more likely to be correct. For example, early exit 1 in Figure 2, with a confidence threshold 20%, produces a hit rate of 35.24% but also a drop of 8.99% in the final accuracy. However, with a confidence threshold 60%, it produces a hit rate 4% and does not reduce the final accuracy by more than 0.1%.

The third observation is that the early exit accuracy is higher than the GT accuracy in all cases. This difference is because we have trained the early exit models to mimic the backbone only by observing its activation values in the intermediate layers and outputs. Since we have not assumed access to the GT labels (which is the case for inference data collected at run-time) while training the early exit models, they have learned to make correct predictions only through predicting the backbone's output, which might have been incorrect in the first place. On the other hand, we observed that the early exit models predict the correct labels for a portion of the samples for which the backbone misclassifies. For example, for 0.92% of the samples, early exit 3 (in Figure 2) correctly predicted the GT labels while the backbone failed ($\overline{B}C$ predictions). This shows the potential of the early exit models to partially compensate for their incorrect early exits ($B\overline{C}$ predictions) by correcting the backbone predictions for some samples ($\overline{B}C$). This agrees with the overthinking concept in SDN (as discussed in 2.3), since for this set of samples, the early exit models have been able to predict correctly in the shallower layers of the backbone.

Figure 2: Individual accuracy and hit rate of the early exit models vs. confidence threshold per early exit model in CIFAR100 - Resnet50 experiment

### 4.6.3 Early Exit models' collaborative performance

Table 2 describes the collaborative performance of the early exit models within a confidence threshold of 0.9. In the table, we also report how each early exit model's layer hits have affected the final accuracy.

Here, we observe that while evaluating the early exit models on the subset of samples, which were missed by the previous early exit models (the relatively more complex ones), the measured hit rate and GT accuracy are substantially lower compared to the evaluation on the whole dataset. This is in fact due to the fact that the simpler samples (less detailed and easier to classify) are resolved earlier in the model. More specifically, the hit rate decreases since the early exit models are less confident in their prediction for the more complex samples, and the GT accuracy also decreases since the backbone is also less accurate for such samples. However, we observe that the early exit models still have high early exit accuracy, with a low impact on the overall accuracy. This observation shows how the confidence-based early exiting method has effectively enabled the early exit models to provide early predictions and keep the overall accuracy drop within the given tolerance.

Table 2: Early Exit models' collaborative performance in terms of hit rate(HR), early exit accuracy ($A_{\text{early exit}}$), GT accuracy ($A_{\text{GT}}$), and their effect on the final accuracy($\downarrow A_{\text{effect}}$) in the early exit-enabled model with a confidence threshold of 0.9. LFW: Labeled Faces in the Wild, MFN: MobileFaceNet, EFN: EfficientNet

| Data | Model | Final accuracy Base | Early Exit-enabled | Exit# | HR | $A_{\text{early exit}}$ | $A_{\text{GT}}$ | $\downarrow A_{\text{effect}}$ |
|---|---|---|---|---|---|---|---|---|
| CIFAR10 | Resnet18 | 88.71% | 86.49% | 1 | 67.21% | 92.29% | 88.91% | 01.31% |
| | | | | 2 | 10.33% | 89.76% | 76.63% | 0.56% |
| | | | | 3 | 11.24% | 85.71% | 51.43% | 0.25% |
| | | | | 4 | 8.32% | 91.37% | 35.71% | 0.1 % |
| | Resnet50 | 87.92% | 85.88% | 1 | 61.41% | 89.12% | 86.19% | 1.12% |
| | | | | 2 | 15.73% | 93.01% | 77.84% | 0.58% |
| | | | | 3 | 10.29% | 82.22% | 53.33% | 0.3% |
| | | | | 4 | 6.1% | 97.47% | 42.65% | 0.04% |
| CIFAR100 | Resnet18 | 75.92% | 74.47% | 1 | 11.96% | 99.29% | 82.11% | 0.94% |
| | | | | 2 | 58.26% | 99.62% | 85.41% | 0.1% |
| | | | | 3 | 7.26 % | 93.81% | 59.29% | 0.3% |
| | | | | 4 | 5.36% | 55.56% | 38.89% | 0.11% |
| | Resnet50 | 78.98% | 77.04% | 1 | 11.92% | 76.34% | 80.2% | 1.32% |
| | | | | 2 | 61.98% | 98.56% | 84.55% | 0.34% |
| | | | | 3 | 11.5% | 97.85% | 63.69% | 0.27% |
| | | | | 4 | 7.38% | 73.68% | 52.63% | 0.1% |
| ImageNet | Resnet18 | 69.76% | 68.12% | 1 | 8.21% | 93.55% | 76.42% | 0.59% |
| | | | | 2 | 14.09% | 93.24% | 84.23% | 1.1% |
| | | | | 3 | 38.13% | 83.53% | 88.76% | 1.13% |
| | | | | 4 | 8.78% | 75.23% | 79.72% | 0.61% |
| | Resnet50 | 76.13% | 74.09% | 1 | 9.13% | 91.41% | 92.19% | 0.48% |
| | | | | 2 | 13.89% | 84.78% | 81.77% | 0.89% |
| | | | | 3 | 42.65% | 78.58% | 72.03% | 1.37% |
| | | | | 4 | 3.14% | 73.12% | 71.12% | 0.19% |
| LFW | MFN | 97.78% | 96.91% | 1 | 37.35% | 98.63% | 97.88% | 0.51% |
| | | | | 2 | 41.02% | 99.71% | 99.71% | 0% |
| | | | | 3 | 55.95% | 93.44% | 96.18% | 0.24% |
| | EFN | 97.29% | 95.35% | 1 | 63.73% | 96.82% | 96.24% | 1.67% |
| | | | | 2 | 14.52% | 99.12% | 98.76% | 0.02% |
| CityScape | Mask RCNN | 91.0% | 83.4% | 1 | 34.3% | 58.3% | 57.9% | 0.1% |
| | | | | 2 | 36.34% | 79.2% | 79.1% | 0.24% |
| | | | | 3 | 21.12% | 87.31% | 86.16% | 0.81% |
| Criteo | DLRM | 78.88% | 73.54% | 1 | 12.3% | 74.3% | 70.9% | 0.4% |
| | | | | 2 | 14.3% | 69.5% | 69.3% | 0.23% |
| | | | | 3 | 25.41% | 94.31% | 90.70% | 0.51% |

**Summary for RQ1**

Early Exit models show lower hit rates and GT accuracy for complex samples, but maintain overall accuracy thanks to the effective use of a confidence-based early exiting method.

Table 3: Original and early exit-enabled models FLOPs (M:Mega - $10^6$)

| Dataset(input size) | Model | FLOPs Original | FLOPs Early Exit-enabled (0.9 Confidence) | ↓ Ratio | ↓ Accuracy |
|---|---|---|---|---|---|
| CIFAR10($3 \times 32 \times 32$) | Resnet18 | 765M | 414M | 45.88% | 2.51% |
|  | Resnet50 | 1303M | 601M | 53.87% | 2.32% |
| CIFAR100($3 \times 32 \times 32$) | Resnet18 | 766M | 374M | 51.17% | 1.91% |
|  | Resnet50 | 1304M | 547M | 58.05% | 2.46% |
| ImageNet($3 \times 224 \times 224$) | Resnet18 | 2343M | 1673M | 28.6% | 2.35% |
|  | Resnet50 | 2783M | 2020M | 27.4% | 2.68% |
| LFW($3 \times 112 \times 112$) | MobileFaceNet | 474M | 296M | 37.55% | 0.91% |
|  | EfficientNet | 272M | 182M | 33.08% | 1.99% |
| CityScape($3 \times 2048 \times 1024$) | Mask R-CNN | 4950M | 2730M | 44.84% | 8.61% |
| Criteo(39) | DLRM | 153M | 99M | 35.29% | 6.52% |

Table 4: Original and early exit-enabled models memory usage

| Dataset(input size) | Model | Original Model Size | Original LTMA | Early Exit-enabled (0.9 Confidence) Model Size | Early Exit-enabled (0.9 Confidence) LTMA | ↓Accuracy |
|---|---|---|---|---|---|---|
| CIFAR10($3 \times 32 \times 32$) | Resnet18 | 43MB | 102MB | 97MB | 88MB | 2.51% |
|  | Resnet50 | 91MB | 235MB | 243MB | 201MB | 2.32% |
| CIFAR100($3 \times 32 \times 32$) | Resnet18 | 43MB | 104MB | 383MB | 93MB | 1.91% |
|  | Resnet50 | 91MB | 235MB | 552MB | 189MB | 2.46% |
| ImageNet($3 \times 224 \times 224$) | Resnet18 | 43MB | 110MB | 403MB | 101MB | 2.35% |
|  | Resnet50 | 91MB | 237MB | 572MB | 198MB | 2.68% |
| LFW($3 \times 112 \times 112$) | MobileFaceNet | 286MB | 567MB | 350MB | 515MB | 0.91% |
|  | EfficientNet | 147MB | 371MB | 297MB | 349MB | 1.99% |
| CityScape($3 \times 2048 \times 1024$) | Mask R-CNN | 3680MB | 3925MB | 4171MB | 4216MB | 8.61% |
| Criteo(39) | DLRM | 320MB | 450MB | 332MB | 411MB | 6.52% |

### 4.6.4   RQ2. To what extent can early exit-enabling improve compute requirements?

In this RQ, we showcase the amount of computation early exiting can save in terms of FLOPs count and analyze the memory usage of the models, and quantify the accuracy drop incurred during this operation.

Table 3 shows the average amount of FLOPs computed for inference per sample. Here we observe that shrinking the batches proportionally decreases the FLOPs count required for inference.

In addition, Table 4 shows the memory used to load the models (i.e, the model size) and the total LTMA during inference while inferring for the test set. As expected, the size of the early exit-enabled models is larger than the original model in all cases, since they include the backbone and the additional early exit models. However, the decrease in LTMA in all cases shows a reduced number of memory allocations during the inference time. Generally, a lower LTMA indicates smaller tensor dimensions (e.g., batch size, input, and operator dimensions) (Ren et al., 2021). However, in our case, since we do not change either of the dimensions, the lower LTMA is due to avoiding the computations in the remaining layers after early exit hits which require further memory allocations. A noteworthy observation from this table highlights the substantial memory usage of our object detection approach due to the sizeable model employed. This underscores the notion that implementing early exiting for this purpose does not significantly amplify the memory requirements.

Table 5: end-to-end evaluation of early exit-enabled models improvement in average inference latency, batch size = 32, MFN: MobileFaceNet, EFN: EfficientNet

| Dataset | Model | Original latency | | Early Exit-enabled latency | | ↓ Ratio | |
|---|---|---|---|---|---|---|---|
| | | CPU | GPU | CPU | GPU | CPU | GPU |
| CIFAR10 | Resnet18 | 13.4 ms | 1.08 ms | 10.11 ms | 0.98 ms | 24.55% | 10.2% |
| | Resnet50 | 18.73 ms | 1.81 ms | 14.62 ms | 1.51 ms | 31.08% | 16.57% |
| CIFAR100 | Resnet18 | 14.23 ms | 1.39 ms | 9.39 ms | 1.25 ms | 34.01% | 10.08% |
| | Resnet50 | 19.59 ms | 2.05 ms | 9.02 ms | 1.84 ms | **46.08%** | 16.75% |
| ImageNet | Resnet18 | 38.19 ms | 3.26 ms | 30.23 ms | 2.79 ms | 20.84% | 14.42% |
| | Resnet50 | 47.21 ms | 3.49 ms | 38.74 ms | 2.42 ms | 17.94% | 30.65% |
| LFW | MFN | 25.34 ms | 8.22 ms | 16.91 ms | 7.30 ms | 33.23% | 11.19% |
| | EFN | 39.41 ms | 17.63 ms | 27.98 ms | 14.38 ms | 29.01% | 18.44% |
| CityScape | Mask R-CNN | 895 ms | 145.2 ms | 562.3 ms | 108.7 ms | 45.12% | **35.32%** |
| Criteo | DLRM | 9.0 ms | 1.7 ms | 7.67 ms | 1.38 ms | 14.7% | 18.82% |

Although the FLOPs count and memory usage indicate the model's inference computational requirements, the decreased amount of FLOPs and LTMA does not necessarily lead to proportional reduction in the models' inference latency, which we further investigate in the next RQ.

> **Summary for RQ2**
>
> Early exit-enabled models are larger due to additional early exit layers but reduce memory use during inference by avoiding computations after early exit hits, demonstrating efficient memory management without significantly increasing overall memory requirements.

### 4.6.5 RQ3. How much acceleration does early exit enablement provide on the CPU/GPU?

In this RQ, we investigate the end-to-end improvement that early exit enablement offers. The results of this measurement clearly depend on multiple deployment factors, such as the underlying hardware and framework, and, as we discuss later in the section, their asynchronous computation capabilities.

Table (5) shows the average latency for the base models on CPU and GPU, vs. their early exit-enabled counterparts, evaluated on the test set.

The first key observation here is the improvements on the CPU. This improvement is due to the low parallelism in the CPU architecture. Essentially, the CPU computation volume is proportional to the number of samples. Therefore, when a sample has an early exit, the remaining computation required to complete the tasks for the batch decreases proportionally.

The FLOPs of all our early exit models, utilized by each input in the batch, are aggregated, resulting in latency being dictated by the longest time observed in the batch. Consequently, a larger batch size tends to worsen latency, as it may lead to more frequent early exit misses.

The second observation is the relatively lower latency improvement in the GPU. This observation shows that shrinking a batch does not proportionally reduce the inference time on GPU, which is due to the high parallelism in the hardware. Shrinking the batch on GPU provides a certain overhead since it interrupts the on-chip parallelism and hardware optimizations. This interruption forces the hardware to re-plan its computations, which can be time consuming. Thus, batch-scanning improvements can be insignificant on GPU. The third observation pertains to the time savings related to pedestrian detection, in contrast to the primary model. This significant gain in efficiency is attributed to disregarding the additional layers of Mask R-CNN through our early exit strategy.

Table 6 further demonstrates how the batch size affects the improvement provided by early exiting. The key observation here is that increasing the batch size can negate the early exiting effect on the inference latency,

Table 6: Inference latency improvement on GPU vs. batch size in Resnet18 and Resnet50 trained on CIFAR100

| Model | Batch Size | Original Latency | Early Exit-enabled (0.9 Confidence) Latency | ↓ Ratio |
|---|---|---|---|---|
| Resnet18 | 16 | 1.34 ms | 1.18 ms | 11.83% |
| | 32 | 1.39 ms | 1.25 ms | 10.08% |
| | 64 | 1.43 ms | 1.77 ms | -24.28% |
| | 128 | 1.61 ms | 2.11 ms | -31.05% |
| Resnet50 | 16 | 1.98 ms | 1.71 ms | 13.68% |
| | 32 | 2.05 ms | 1.84 ms | 16.75% |
| | 64 | 2.19 ms | 1.98 ms | 9.21% |
| | 128 | 2.7 ms | 3.22 ms | -19.43% |

which, as discussed, is due to fewer batches that are fully resolved through the early exit models and do not reach the last layers. In conclusion, the latency improvement here highly depends on the hardware used in inference and must be specifically analyzed per hardware environment and computation parameters such as batch size. However, the method can still be useful when the model is not performing batch inferences (batch size = 1). One can also use the tool and get the best prediction so far within the forward-pass process by disabling batch shrinking. Doing so will generate multiple predictions per input sample, one per exit (early and final).

> **Summary for RQ3**
>
> Increasing batch size in early exit models can worsen latency due to more frequent early exit misses and does not proportionally reduce inference time on GPUs due to the disruption of hardware parallelism. In particular, significant time savings are achieved in pedestrian detection by bypassing extra layers with an early exit strategy, although the benefits of early exiting on latency heavily depend on the specific hardware and batch size used.

### 4.6.6 RQ4. How does the early exit-enabled model accuracy / latency and computational cost trade-off compare with other early exit methods?

In this RQ, we conducted a comprehensive evaluation of three distinct methods, including BranchyNet (Teerapittayanon et al., 2016), GATI (Balasubramanian et al., 2021), and our proposed method in the same system configuration. This evaluation included two different Resnets with two different data sets to thoroughly assess the performance of each method.

Figures 3 and 4 demonstrate a comprehensive analysis of the accuracy / latency / computational cost trade-off at various confidence levels for all methods evaluated for CIFAR10 and CIFAR100. Our findings reveal a remarkable adaptability of our proposed method to different confidence levels, particularly excelling in low-confidence scenarios. This adaptability underscores the effectiveness of our approach's training, showcasing its ability to perform exceptionally well in situations where traditional methods might falter. In contrast, as confidence levels increase, our method exhibits a slight latency increase, reflecting its ability to fine-tune confidence settings to suit various application requirements. This feature demonstrates the method's adaptability and its potential for accommodating diverse use cases.

Table 7 presents the comparison of inference costs and accuracy across different methods (BranchyNet, GATI, and SelfXit ) using ResNet18 and ResNet50 models on CIFAR-10 and CIFAR-100 datasets. The table reports accuracy at various inference cost thresholds ($\leq 25\%$, $\leq 50\%$, and $\leq 75\%$) relative to the original model's inference cost. The "Max" column indicates the highest accuracy achieved by each method with early exits. This comparison highlights the trade-offs between inference cost and accuracy for the evaluated methods. This table demonstrates that our method has a better impact for more complex base models and datasets.

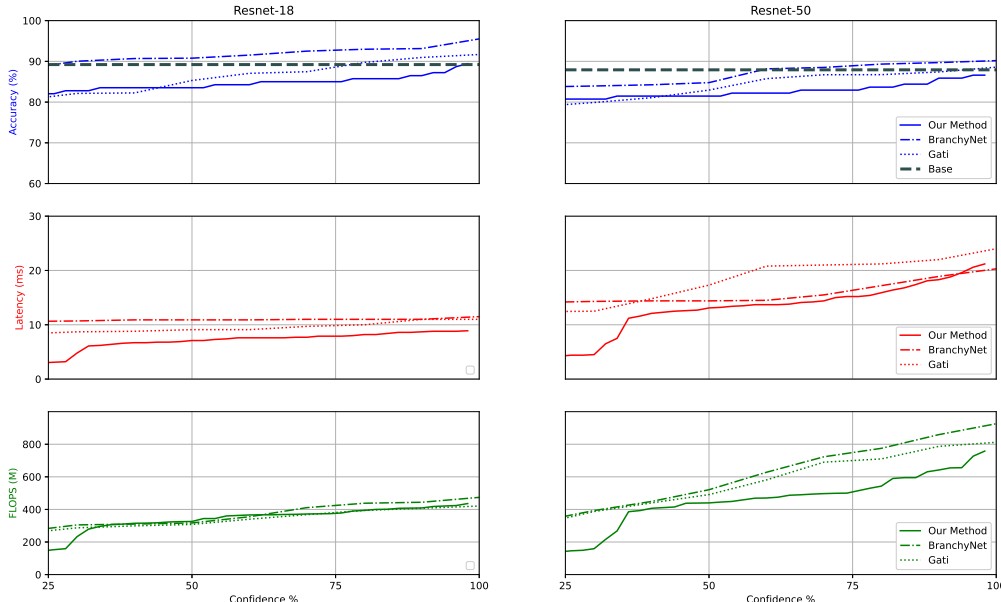

Figure 3: CIFAR10 Accuracy/latency/computational cost comparison between the approaches with different confidence.

Table 7: Comparing the inference costs of Resnet18 and Resnet50 models on CIFAR-10 and CIFAR-100 on different methods. $\leq 25\%$, $\leq 50\%$, and $\leq 75\%$ report the early exit accuracy when we limit the average inference cost to at most 25%, 50%, and 75% that of the original models of each method. Max reports the highest accuracy early exits can achieve.

| Method | $\leq 25\%$ | $\leq 50\%$ | $\leq 75\%$ | Max |
|---|---|---|---|---|
| CIFAR-10 - ResNet18 | | | | |
| BranchyNet | 86.4 | 87.7 | 91.54 | 95.5 |
| GATI | 69.8 | 77.4 | 85.7 | 91.6 |
| SelfXit | 81.6 | 82.8 | 83.5 | 89.4 |
| CIFAR-10 - ResNet50 | | | | |
| BranchyNet | 82.0 | 84.3 | 88.4 | 90.1 |
| GATI | 74.5 | 80.3 | 86.0 | 88.5 |
| SelfXit | 80.7 | 81.5 | 83.7 | 86.6 |
| CIFAR-100 - ResNet18 | | | | |
| BranchyNet | 70.8 | 72.8 | 73.2 | 77.4 |
| GATI | 71.5 | 72.5 | 74.7 | 76.4 |
| SelfXit | 62.2 | 63.0 | 67.8 | 77.5 |
| CIFAR-100 - ResNet50 | | | | |
| BranchyNet | 65.7 | 69.5 | 72.3 | 74.1 |
| GATI | 66.2 | 66.8 | 69.7 | 73.3 |
| SelfXit | 67.6 | 70.0 | 73.9 | 79.4 |

The primary observation is that our method exhibits the best latencies in various methods at different confidence levels. Another significant observation is that our method's performance scales positively with the complexity of the model or dataset, leading to superior outcomes in terms of both accuracy and latency. As we add some lightweight layers for each model, it becomes evident that Resnet50 exhibits more pronounced

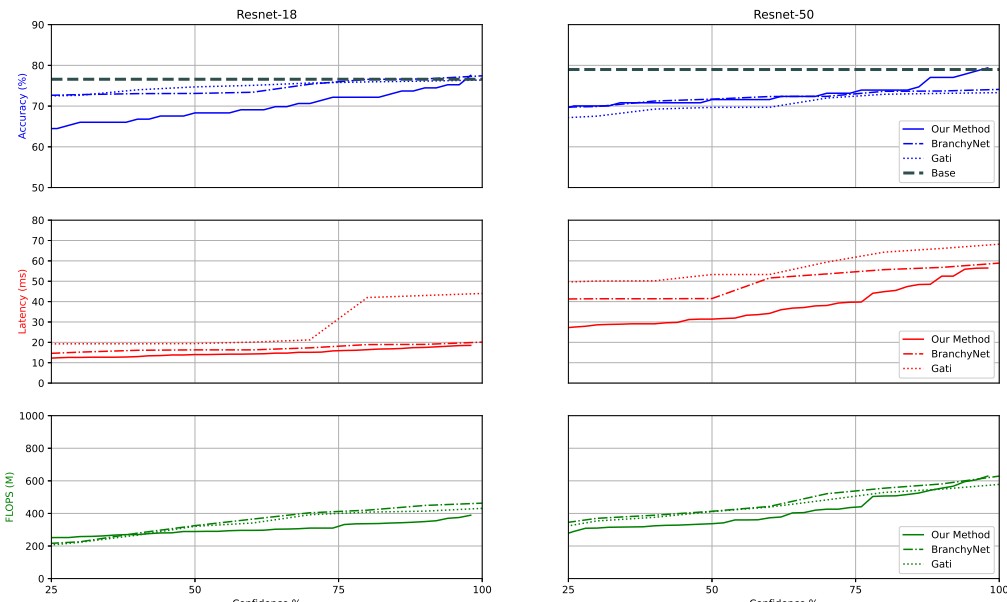

Figure 4: CIFAR100 Accuracy/latency/computational cost comparison between the approaches with different confidence.

improvements in our research. This observation is also shown in Figure 3. By averaging across all confidence levels greater than 0.25, For the simplest model and dataset, namely CIFAR10-Resnet18, we achieved a substantial 52.2% and 32.4% reduction in latencies, with a modest 6.9% and 1.6% decrease in accuracy when compared to the BranchyNet and GATI methods, respectively, while improving computational costs by 7.1% and 0.5%. In the case of a more complex model and dataset, such as CIFAR100-Resnet50, we achieved a substantial reduction in latency of 51.6% and 30.4% while simultaneously improving the accuracy by 2.31% and 0.72% compared to the GATI and BranchyNet methods, respectively, while improving computational costs by 14.3% and 8.2%. BranchyNet, on the other hand, exhibits relatively consistent results at different confidence levels. Although this stability might be desirable in some scenarios, it lacks the adaptability and dynamic response that our method offers. Gati, while showing learned early exits, is marked by the complexity of its training process, which can be challenging to implement effectively. Moreover, GATI consumes more memory resources during implementation, in contrast to the efficiency of SelfXit .

Indeed, our method stands out as a dynamic and adaptable solution, requiring less extensive data and model preparation compared to approaches such as GATI. By completely freezing the base model during the training of early-exit layers, we achieve better results within limited training time, particularly in more complex models where other methods may require significantly more time to train effectively. The ability to configure confidence levels further underscores its versatility for a broad spectrum of applications. However, it should be noted that for simpler models and datasets, our method may not demonstrate a marked improvement over others, as they employ additional dense layers for early exiting, which can provide competitive results. Crucially, the most significant aspect of our work lies in the capability to update the early exit layers effectively without reliance on ground-truth labels, aligning early-exit results closely with the base model's predictions. This offers a substantial advantage in practical applications where such labels may not be readily available.

**Summary for RQ4**

Our method demonstrates exceptional adaptability across different confidence levels, particularly excelling in low-confidence scenarios, which highlights its robust training and ability to fine-tune settings for varied applications. It achieves the best latencies and scales positively with the complexity of the model or dataset, showing substantial improvements in both accuracy and computational costs, particularly with complex configurations like CIFAR100-Resnet50.

To evaluate the impact of implementing NAS and varying model depths, we conducted an ablation study, detailed in Appendix D.

### 4.7 Discussion, Limitations, and Future Directions

The first limitation of this study is that the proposed method is limited to classification models, since it would be more complicated for early exit models to predict the output of a regression model due to their continuous values. This limitation is strongly tied to the effectiveness of knowledge distillation in the case of regression models.

The method also does not take the internal state of the backbone (if any) into account, such as the hidden states in recurrent neural networks. Therefore, the effectiveness of the method for such models still needs to be further assessed.

Moreover, practitioners should take the underlying hardware and the backbone structure into account, as they directly affect the final performance. On this note, as shown in Section 4.6.5, different models provide different performances in terms of inference latency in the first place; therefore, choosing the right model for the task comes first, and early exiting can be helpful in improving the performance.

As the main goal of our proposed approach is to decrease the inference time, depending on the application domain, if a potential slight degradation of accuracy is not acceptable (e.g., in a safety critical system), using our approach might not be a good fit.

In dynamic environments where data distribution changes over time, maintaining the performance of early exit models may require periodic updates. Unlike conventional caching, early exit models cannot be updated in real-time. Therefore, retraining the early exit layers and readjusting confidence thresholds using newly collected inference samples become necessary to adapt to new data trends and maintain accuracy. While we did not perform these updates in our experiments, practitioners deploying this method should consider triggers for updates, such as when a significant portion of new data has been collected or when the backbone model is modified or retrained. Balancing the costs associated with retraining against application requirements and resources is essential for effective maintenance.

While our current comparison has yielded valuable results, we can explore the applicability of our approach to other large models, particularly in non-vision-based datasets, to assess its effectiveness in different domains. Given the growing importance of reducing latency and inference time, especially in Large Language Models (LLMs), our future research can focus on methods to further optimize and reduce costs for large-scale industries.

## 5 Conclusion

In this paper, we have shown that our automated early exiting approach is able to extend a pretrained classification DNN to an early exit-enabled version using a relatively small and unlabeled dataset. The required training data sets for early exiting models are collected just by recording the input items and their corresponding backbone outputs at the inference time. We have also shown that the early exiting method can introduce a significant improvement in the model's computing requirements and inference latency, especially when the inference is performed on the CPU.

We discussed the parameters, design choices, and the procedure of early exit-enabling a pre-trained off-the-shelf model.

In conclusion, while traditional early exiting might not be beneficial for DNN models due to the diversity, size, and dimensions of the inputs, early exiting the features in the hidden layers of the DNNs using the early exit models can achieve significant improvement in the model's inference computational complexity and latency. As shown in sections 4.6.4 and 4.6.5, early exiting reduces the average inference FLOPs by up to 58% and the latency by up to 46.09% on CPU and 18.44% on GPU for classification purposes. For pedestrian detection, we could reduce latency up to 45.1% on CPU and 35.32% on GPU. In summary, SelfXit consistently outperforms alternative approaches in different models and datasets. Considering average values across all confidence levels greater than 0.25, in the case of the simplest model, CIFAR10-Resnet50, we observe a remarkable reduction of 52.2% and 32.4% in latency, with accuracy only experiencing a minor decrease of 6.9% and 1.6% compared to the BranchyNet and GATI methods, respectively. For the more intricate CIFAR100-Resnet50 model and dataset, our method excels with a significant 51.6% and 30.4% reduction in latency, and a notable 2.31% and 0.72% improvement in accuracy compared to the BranchyNet and GATI methods. These findings underscore the adaptability and superior performance of our method in diverse and complex scenarios.

## Acknowledgments

This work was partially supported by NSERC (ALLRP/568643-2021), Alberta Innovates (212200865) Alliance Grants, and National Science Foundation (Awards 2007202, 2107463, 2038080, and 2233873), and the Natural Sciences and Engineering Research Council of Canada (NSERC) Discovery Grant Program (RGPIN-2024-03954).

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

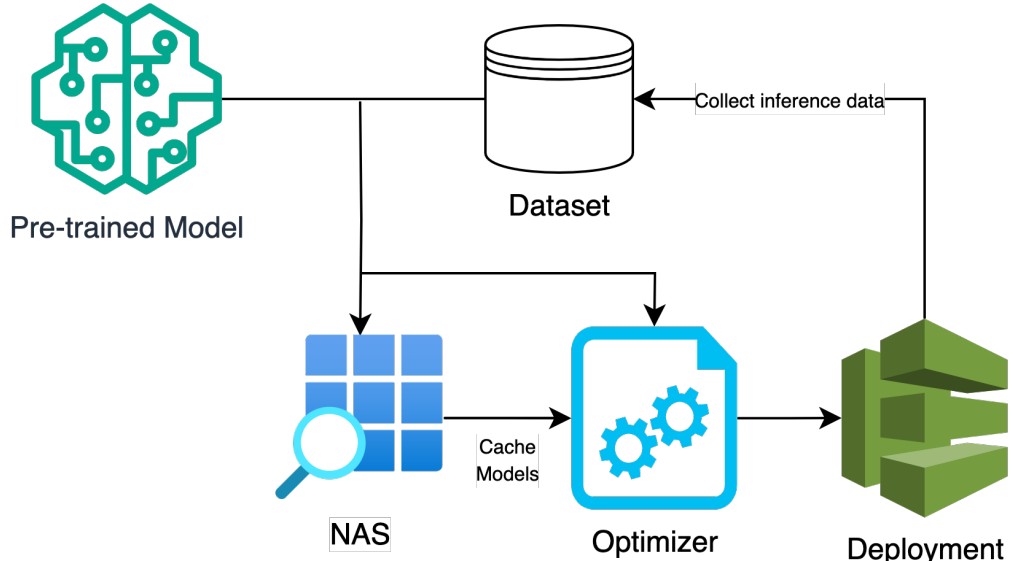

Figure 5: Early Exiting system overall framework

# A Appendix: Implementation

## A.1 Implementation details

We developed the early exit tool using PyTorch, which is accessible through the GitHub repository[2]. The software environment for our experiments included Ubuntu 20.04, Python 3.7, and PyTorch 1.1. Figure 5 shows the overall system design. The tool provides a NAS module, an optimizer module, and a deployment module. The NAS module provides the architectures to be used per early exit model. The optimizer assigns the confidence thresholds, finds the best subset of the early exit models, and provides evaluation reports. Lastly, the deployment module launches a Web server with the early exit-enabled model ready to serve queries.

### A.1.1 NAS Module

Existing NAS tools typically define different search spaces according to different tasks, which constrains their applicability to certain input types and sizes. Using such tools with input constraints defeats our method's generalization and automation purpose, since the early exit models' inputs can have any dimension and size. For example, ProxylessNAS (Cai et al., 2019) specializes in optimizing the performance of neural architecture for a target hardware. However, it is only applicable for image classification tasks and requires certain input specifications (e.g., 3xHxW images normalized using given values). Similarly, Auto-PyTorch (Zimmer et al., 2021) and Auto-Keras are only applicable to tabular data sets, text, and images.

We chose NNI from Microsoft (Microsoft, 2021) as it does not constrain the input of the model in terms of type, size, and dimensions. NNI also provides an extensible search space definition with support for variable number of layers and nested choices (e.g., choosing among different layer types, each with different layer-specific parameters).

Given the backbone implementation, the dataset and the search space, the module launches an NNI experiment per candidate layer to find an optimum early exit model for the layer. Each experiment launches a web GUI for progress reports and results.

---

[2]https://github.com/hoseinkhs/AutoCacheLayer/

We aim for end-to-end automation in the tool. However, currently, the user still needs to manually export the architecture specifications when using the NAS module and convert them to a proper Python implementation (i.e., a PyTorch module implementing the architecture). The specifications are available to the user through the experiments web GUI and also in the trial output files. This shortcoming is due to the NNI implementation, which does not currently provide access to the model objects within the experiments. We have created an enhancement suggestion on the NNI repository to support model object access (issue #4910).

### A.1.2  Optimizer and deployment modules

Given the implementation of the backbone and the early exit models, the optimizer evaluates the early exit models, assigns their confidence thresholds, finds the best subset of the early exit models, and disables the rest, and finally reports the relevant performance metrics for the early exit-enabled model and each early exit model. We used the DeepSpeed by Microsoft and the PyTorch profiler to profile the FLOP counts, memory usage, and latency values for the early exit models and the backbones.

The user can use each module independently. Specifically, the user can skip the architecture search via the NAS module and provide the architectures manually to the optimizer, and the module trains them before proceeding to the evaluation.

NAS with random search strategy can take approximately 1 hour for simpler models like ResNet18 and up to 3 hours for more complex models like ResNet50.

The tool also offers an extensive set of configurations. More specifically, the user can configure the tool to use one device (e.g., GPU) for training processes and the other (e.g., CPU) for evaluation and deployment.

The deployment module launches a web server and exposes a WebSocket API to the early exit-enabled model. The query batches passed to the socket will receive one response per item, as soon as the prediction is available through either of the (early or final) exits.

### A.1.3  Backbone Implementation

We used the backbone implementations and weights provided by the FaceX-Zoo (Wang et al., 2021) repository to conduct the experiments with LWF data set on the MobileFaceNet and EfficientNet models.

For experimenting with CIFAR10 and CIFAR100, we used the implementations provided by torchvision (Marcel & Rodriguez, 2010) and the weights provided by (Phan, 2021) and (Weiaicunzai, 2020). For ImageNet pre-trained weights for ResNets we used new torchvision version as well (maintainers & contributors, 2016). All backbone implementations were modified to implement an interface that handles the interactions with the early exit models, controls the exits (early exit hits and misses), and provides the relevant reports and metrics. We document the use of the interface in the repository, so that users can experiment with new backbones and datasets. We refer interested readers to a blog post on how to extract intermediate activations in PyTorch (Bhaskhar, 2020) which introduces three methods to access the activation values. The introduced forward hooks method in PyTorch is very convenient for read-only purposes. However, our method requires performing actions based on the activation values, specifically, early exit lookup and batch shrinking, and avoiding further computation through the next layers. Therefore, we used the so-called "hacker" method to access the activation values and perform these actions and provided the interface for easy replication on different backbones.

### A.2  Environment setup

The hardware used for inference substantially affects the results due to the hardware-specific optimizations such as computation parallelism. In our experiments, we have used an "Intel(R) Core(TM) i7-10700K CPU @ 3.80GH" to measure on-CPU inference times and an "NVIDIA GeForce RTX 3070" GPU to measure on-GPU inference time.

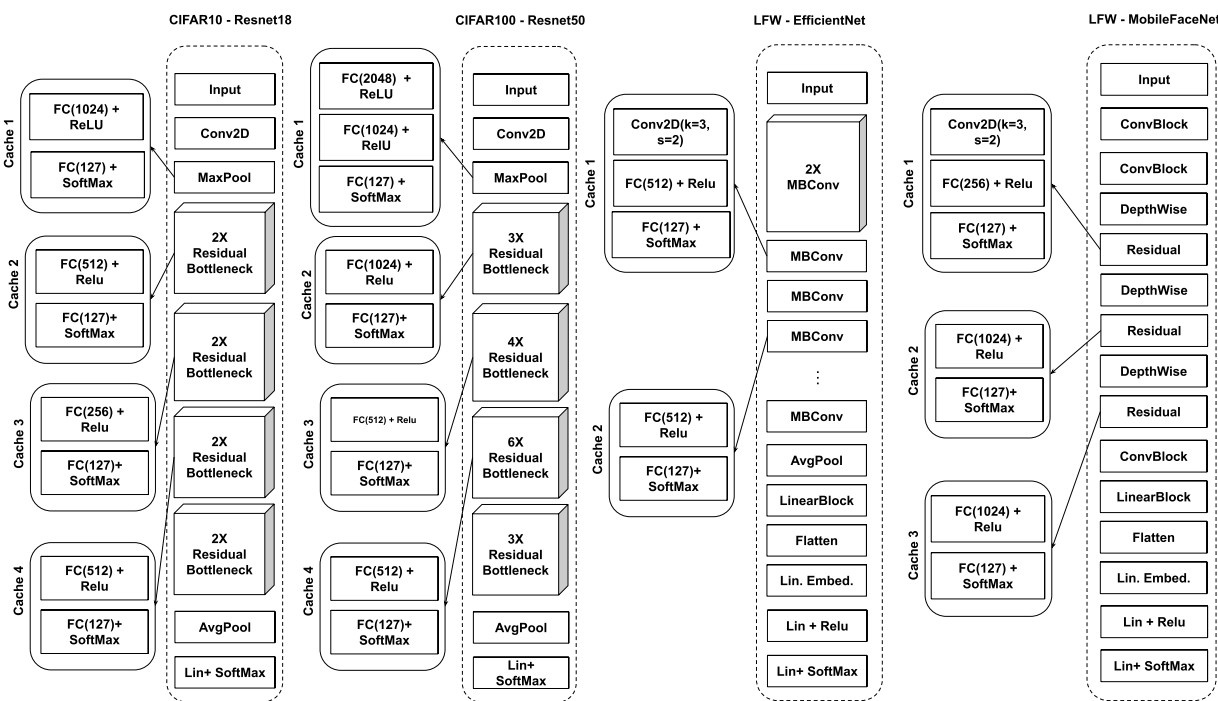

Figure 6: Final schema of the early exit models, for the experiments CIFAR10-Resnet18, CIFAR100-Resnet50, LWF-EfficientNet, and LFW-MobileFaceNet

## B    Appendix: Final schema of early exit models

This appendix shows key figures that provide a deeper visual understanding of the methodologies and results highlighted in the main text.

Figure 6 illustrates the final schema of the early exit models used in our experiments with CIFAR10-Resnet18, CIFAR100-Resnet50, LWF-EfficientNet, and LFW-MobileFaceNet.

Figure 7 presents the final schema of the early exit models specifically designed for our CityScapes experiments using the Mask R-CNN framework.

Figure 8 shows the final schema of the early exit models designed for our recommendation experiments with the Criteto data set using DLRM.

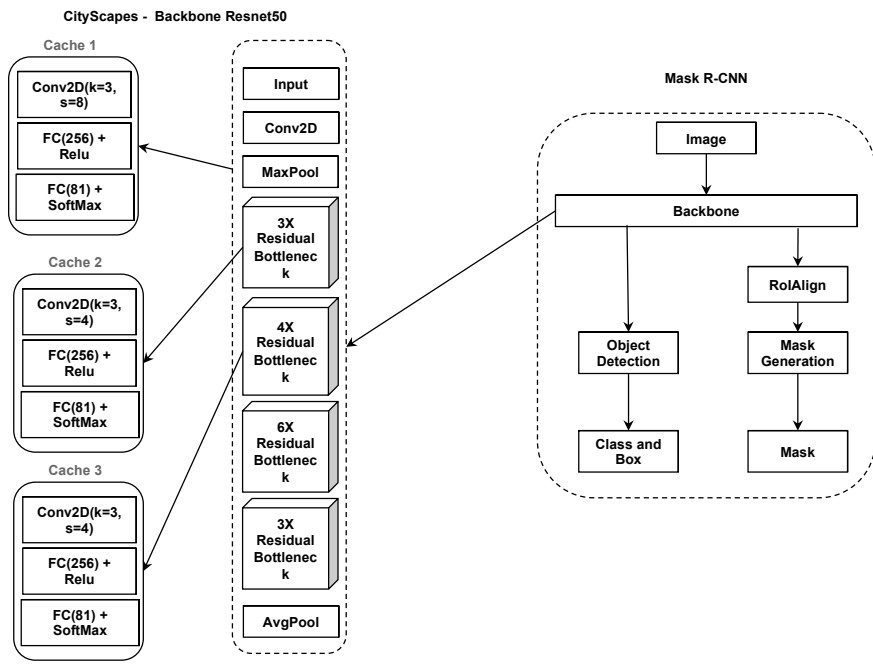

Figure 7: Final schema of the early exit models, for the experiments CityScapes - Mask RCNN

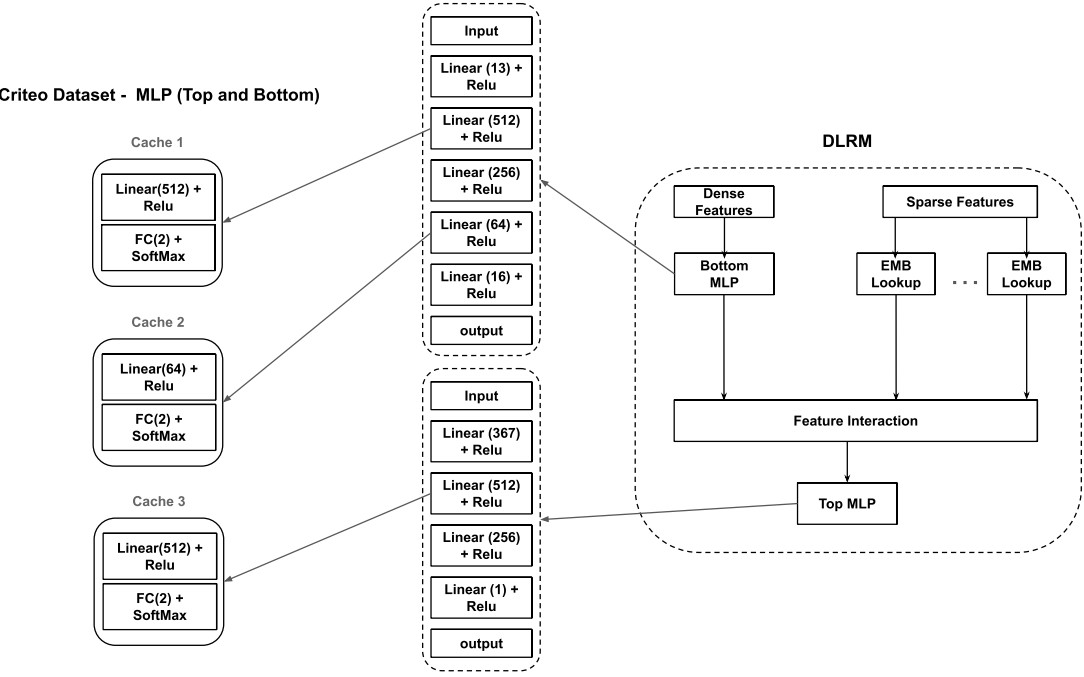

Figure 8: Final schema of the early exit models, for the experiments Criteo - DLRM

## C   Appendix: Early Exit models' individual performance for all experiments

The following figures demonstrate the hit rate, GT accuracy and early exit accuracy of each early exit model versus the confidence threshold, per experiment dataset and backbone.

Figure 9: Experiment: CIFAR10-Resnet18

Table 8: Different early exit models accuracy, MFN: MobileFaceNet, EFN: EfficientNet, R18: Resnet18, R50: Resnet50, C10: CIFAR10, C100: CIFAR100, IN: ImageNet, CityScapes: CS, Mask RCNN: MRCNN

| Model Type | R18-C10 | R50-C10 | R18-C100 | R50-C100 | R18-IN | R50-IN | MFN-LFW | EFN-LFW | MRCNN-CS | DLRM-Criteo |
|---|---|---|---|---|---|---|---|---|---|---|
| With NAS | 86.49% | 85.88% | 74.47% | 77.04% | 68.12% | 74.09% | 96.91% | 95.35% | 83.4% | 73.54% |
| Without NAS | 86.79% | 85.16% | 74.32% | 77.19% | 67.31% | 72.91% | 96.19% | 95.71% | 83.14% | 73.96% |
| 1 more layer | 87.15% | 86.12% | 74.29% | 77.72% | 67.56% | 73.19% | 96.07% | 95.92% | 83.89% | 73.91% |
| 2 more layers | 86.20% | 86.09% | 74.82% | 77.02% | 67.61% | 73.21% | 95.42% | 95.14% | 83.01% | 73.62% |
| 3 more layers | 86.17% | 86.01% | 74.78% | 77.03% | 67.60% | 73.31% | 95.43% | 95.14% | 82.99% | 73.48% |

# D  Appendix: Ablation Study

To investigate the need for using NAS in our work, we run an ablation study. The alternative to use NAS is inserting early exit models at every possible node, which can be memory-intensive and may even degrade the final performance and accuracy. Additionally, the number of early exit layers may also impact the results. Our default setup uses 2 dense layers per early exit model. This part of the ablation study assesses the impact of increasing this number to 3, 4, and 5 dense layers. The following tables compare latency (GPU and CPU), accuracy, model size and FLOPs across these modes.

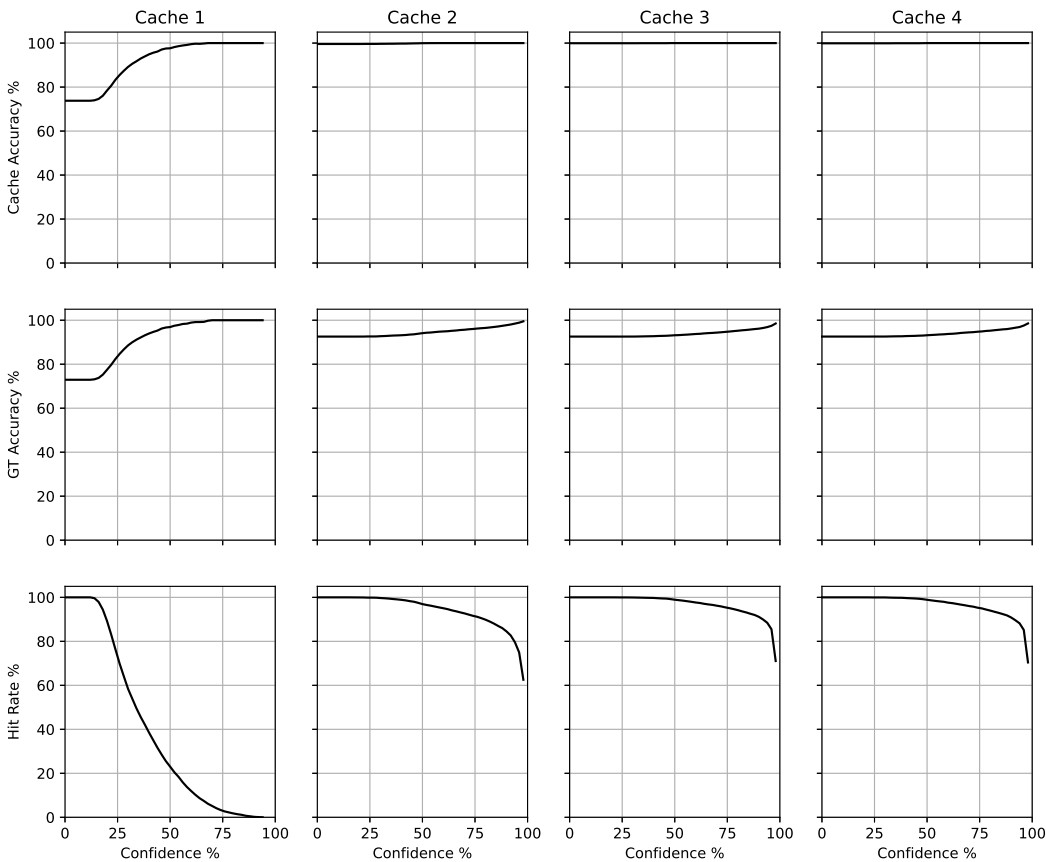

Figure 10: Experiment: CIFAR10-Resnet50

Table 9: Different early exit models latency on GPU, MFN: MobileFaceNet, EFN: EfficientNet, R18: Resnet18, R50: Resnet50, C10: CIFAR10, C100: CIFAR100, IN: ImageNet, CityScapes: CS, Mask RCNN: MRCNN

| Model Type | R18-C10 | R50-C10 | R18-C100 | R50-C100 | R18-IN | R50-IN | MFN-LFW | EFN-LFW | MRCNN-CS | DLRM-Criteo |
|---|---|---|---|---|---|---|---|---|---|---|
| With NAS | 0.98 ms | 1.51 ms | 1.25 ms | 1.84 ms | 2.79 ms | 2.42 ms | 7.30 ms | 14.38 ms | 108.7 ms | 1.38 ms |
| Without NAS | 1.29 ms | 1.75 ms | 1.47 ms | 2.27 ms | 3.91 ms | 3.64 ms | 9.26 ms | 17.87 ms | 153.1 ms | 1.89 ms |
| 1 more layer | 1.11 ms | 1.63 ms | 1.32 ms | 2.19 ms | 3.27 ms | 3.33 ms | 7.91 ms | 16.10 ms | 123.1 ms | 1.52 ms |
| 2 more layers | 1.21 ms | 1.71 ms | 1.39 ms | 2.26 ms | 3.34 ms | 3.37 ms | 7.97 ms | 16.70 ms | 131.27 ms | 1.69 ms |
| 3 more layers | 1.27 ms | 1.81 ms | 1.48 ms | 2.37 ms | 3.61 ms | 3.47 ms | 8.83 ms | 17.51 ms | 138.27 ms | 1.74 ms |

Regarding NAS use, the results indicate that there is an average increase in accuracy of less than 2%, while latency, FLOPs, and model sizes increased by an average of 16%. When using additional dense layers, the results showed an increase in accuracy of less than 1%, but resulted in approximately 24% worse results in latency and model size and about 30% increase in FLOPs usage on average.

Ablation experiments indicate that using additional dense layers or ignoring NAS and inserting and training all possible early exit layers may result in slightly better accuracy. However, this is accompanied by increased latency and, in some cases, performance is worse than the original model without early exits. This is because

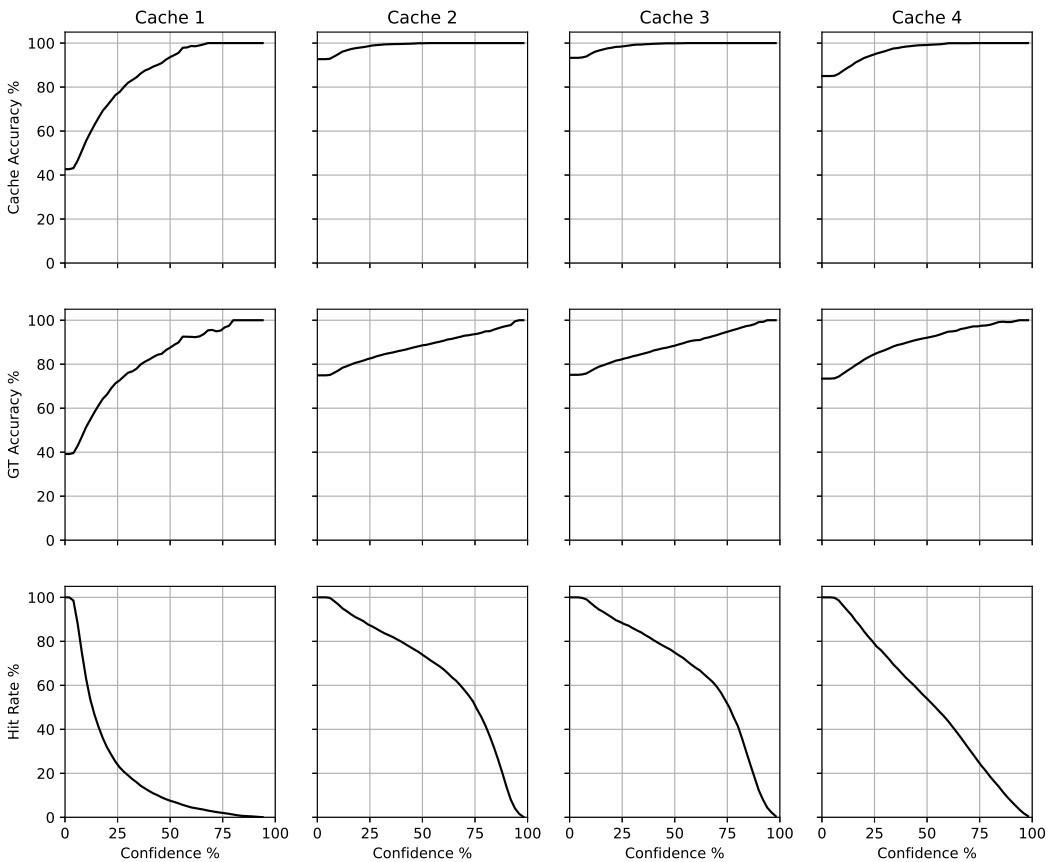

Figure 11: Experiment: CIFAR100-Resnet18

Table 10: Different early exit models latency on CPU, MFN: MobileFaceNet, EFN: EfficientNet, R18: Resnet18, R50: Resnet50, C10: CIFAR10, C100: CIFAR100, IN: ImageNet, CityScapes: CS, Mask RCNN: MRCNN

| Model Type | R18-C10 | R50-C10 | R18-C100 | R50-C100 | R18-IN | R50-IN | MFN-LFW | EFN-LFW | MRCNN-CS | DLRM-Criteo |
|---|---|---|---|---|---|---|---|---|---|---|
| With NAS | 10.11 ms | 14.62 ms | 9.39 ms | 9.02 ms | 30.23 ms | 38.74 ms | 16.91 ms | 27.98 ms | 562.3 ms | 7.67 ms |
| Without NAS | 13.53 ms | 19.17 ms | 13.19 ms | 13.01 ms | 41.29 ms | 51.04 ms | 22.53 ms | 38.77 ms | 724.42 ms | 11.89 ms |
| 1 more layer | 11.87 ms | 18.12 ms | 12.19 ms | 12.12 ms | 33.31 ms | 45.41 ms | 21.25 ms | 34.68 ms | 689.11 ms | 10.63 ms |
| 2 more layers | 12.84 ms | 19.32 ms | 14.01 ms | 13.89 ms | 34.91 ms | 47.02 ms | 25.94 ms | 35.78 ms | 713.71 ms | 11.93 ms |
| 3 more layers | 13.17 ms | 22.12 ms | 16.71 ms | 15.89 ms | 36.06 ms | 48.68 ms | 26.41 ms | 37.52 ms | 743.71 ms | 13.31 ms |

having, for example, 10 early exit layers and processing all batches through them significantly increases the time required and also enlarges the model size. In addition, memory consumption is negatively affected.

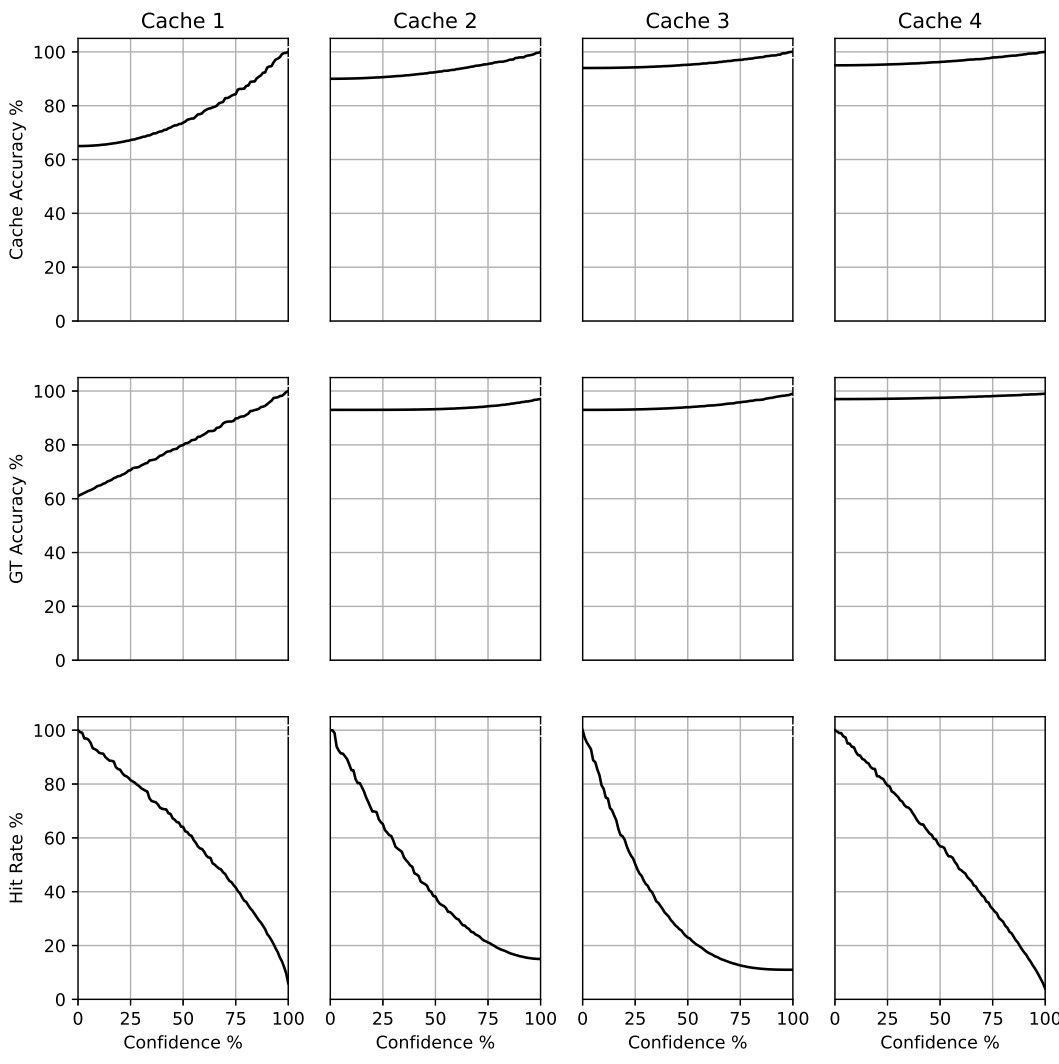

Figure 12: Experiment: ImageNet-Resnet18

Table 11: Different early exit models FLOPs, MFN: MobileFaceNet, EFN: EfficientNet, R18: Resnet18, R50: Resnet50, C10: CIFAR10, C100: CIFAR100, IN: ImageNet, CityScapes: CS, Mask RCNN: MRCNN

| Model Type | R18-C10 | R50-C10 | R18-C100 | R50-C100 | R18-IN | R50-IN | MFN-LFW | EFN-LFW | MRCNN-CS | DLRM-Criteo |
|---|---|---|---|---|---|---|---|---|---|---|
| With NAS | 414M | 601M | 374M | 547M | 1673M | 2020M | 296M | 182M | 2730M | 99M |
| Without NAS | 671M | 942M | 545M | 719M | 1941M | 2430M | 371M | 282M | 3211M | 128M |
| 1 more layer | 614M | 921M | 534M | 643M | 1897M | 2321M | 351M | 261M | 3139M | 121M |
| 2 more layers | 701M | 1011M | 709M | 773M | 1970M | 2410M | 404M | 290M | 3310M | 138M |
| 3 more layers | 791M | 1125M | 771M | 823M | 2018M | 2512M | 514M | 368M | 3589M | 153M |

Figure 13: Experiment: ImageNet-Resnet50

Table 12: Different early exit models size (Mega Byte), MFN: MobileFaceNet, EFN: EfficientNet, R18: Resnet18, R50: Resnet50, C10: CIFAR10, C100: CIFAR100, IN: ImageNet, CityScapes: CS, Mask RCNN: MRCNN

| Model Type | R18-C10 | R50-C10 | R18-C100 | R50-C100 | R18-IN | R50-IN | MFN-LFW | EFN-LFW | MRCNN-CS | DLRM-Criteo |
|---|---|---|---|---|---|---|---|---|---|---|
| With NAS | 97MB | 243MB | 383MB | 552MB | 403MB | 572MB | 350MB | 297MB | 4171MB | 332MB |
| Without NAS | 137MB | 324MB | 442MB | 822MB | 731MB | 801MB | 514MB | 411MB | 5231MB | 451MB |
| 1 more layer | 113MB | 273MB | 401MB | 622MB | 591MB | 609MB | 372MB | 320MB | 4319MB | 362MB |
| 2 more layers | 130MB | 301MB | 420MB | 691MB | 682MB | 725MB | 395MB | 351MB | 4480MB | 390MB |
| 3 more layers | 159MB | 340MB | 451MB | 751MB | 705MB | 789MB | 421MB | 393MB | 4680MB | 427MB |

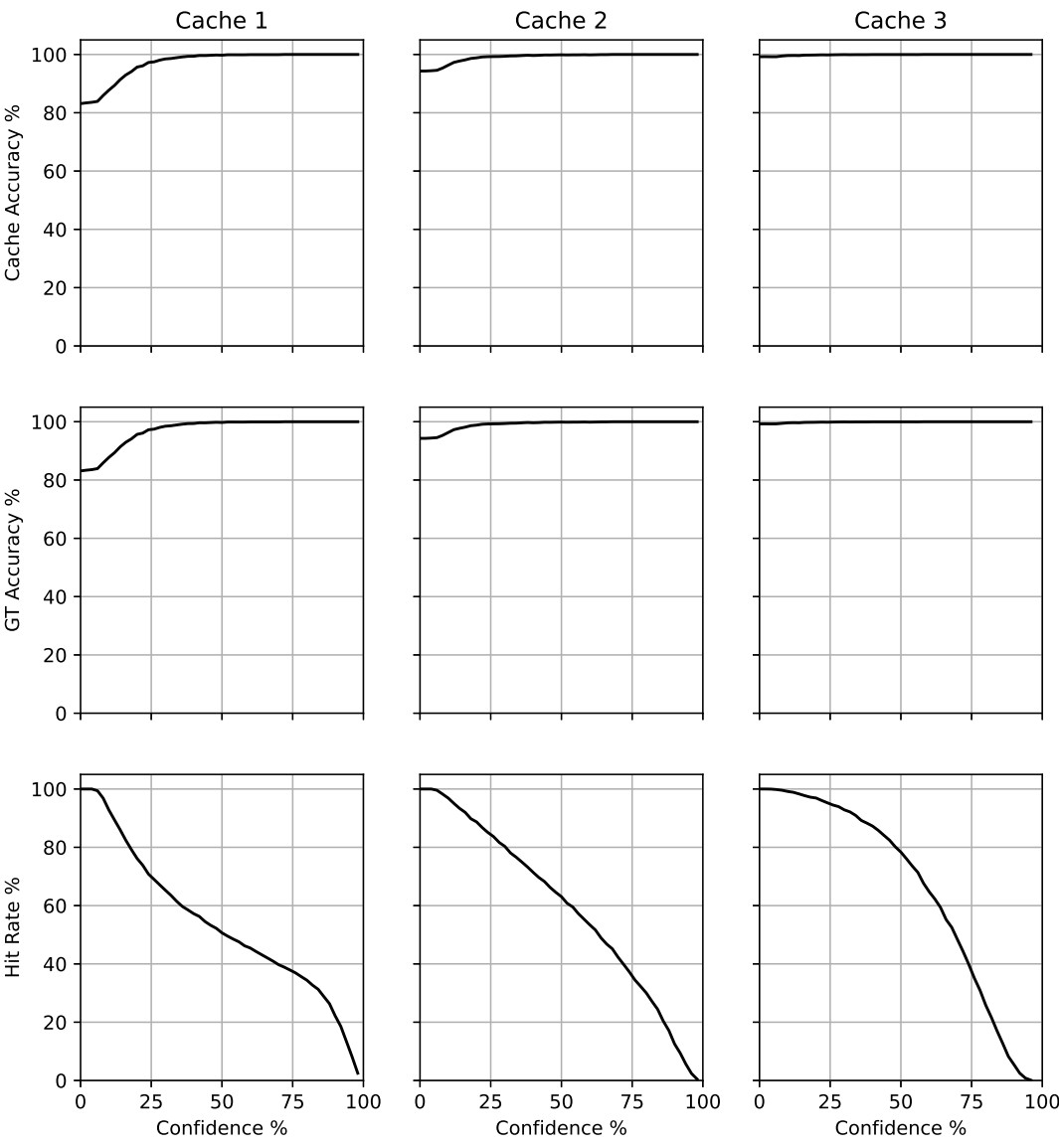

Figure 14: Experiment: LFW-MobileFaceNet

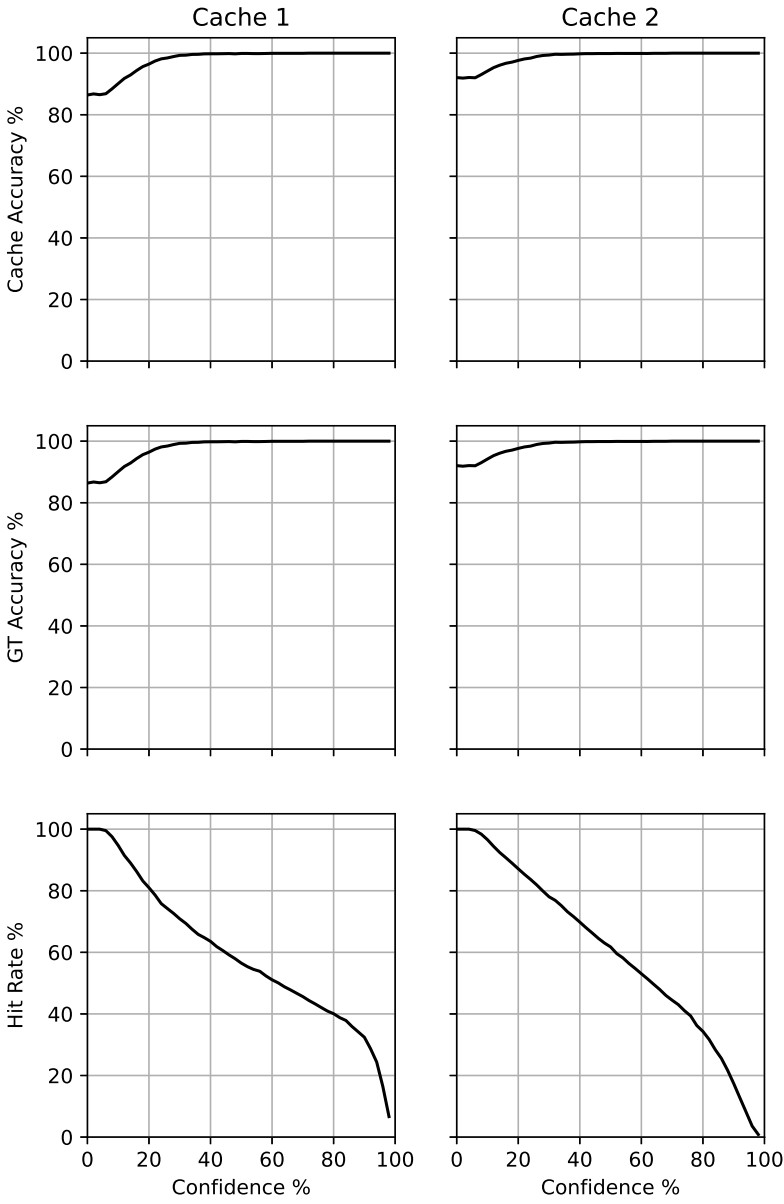

Figure 15: Experiment: LFW-EfficientNet

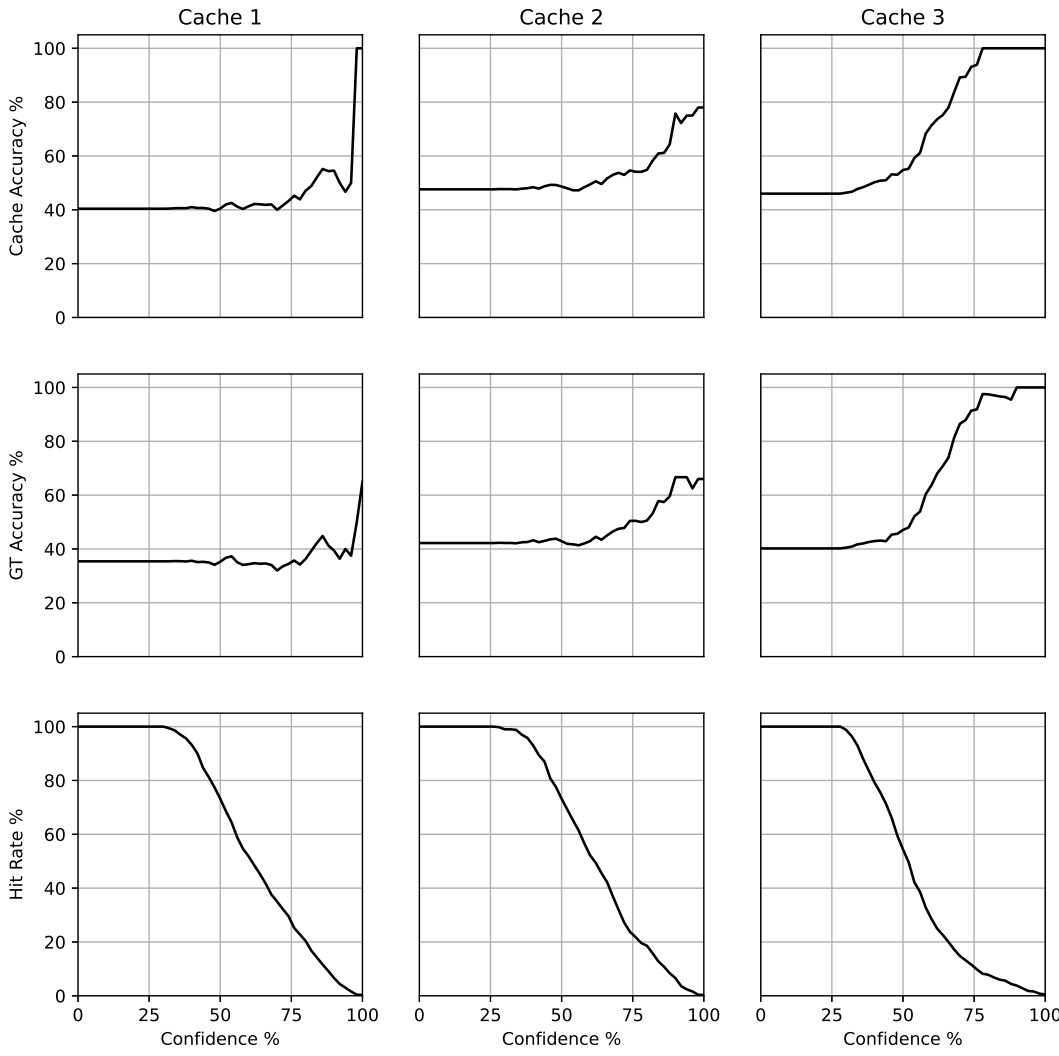

Figure 16: Experiment: Mask RCNN-CityScape

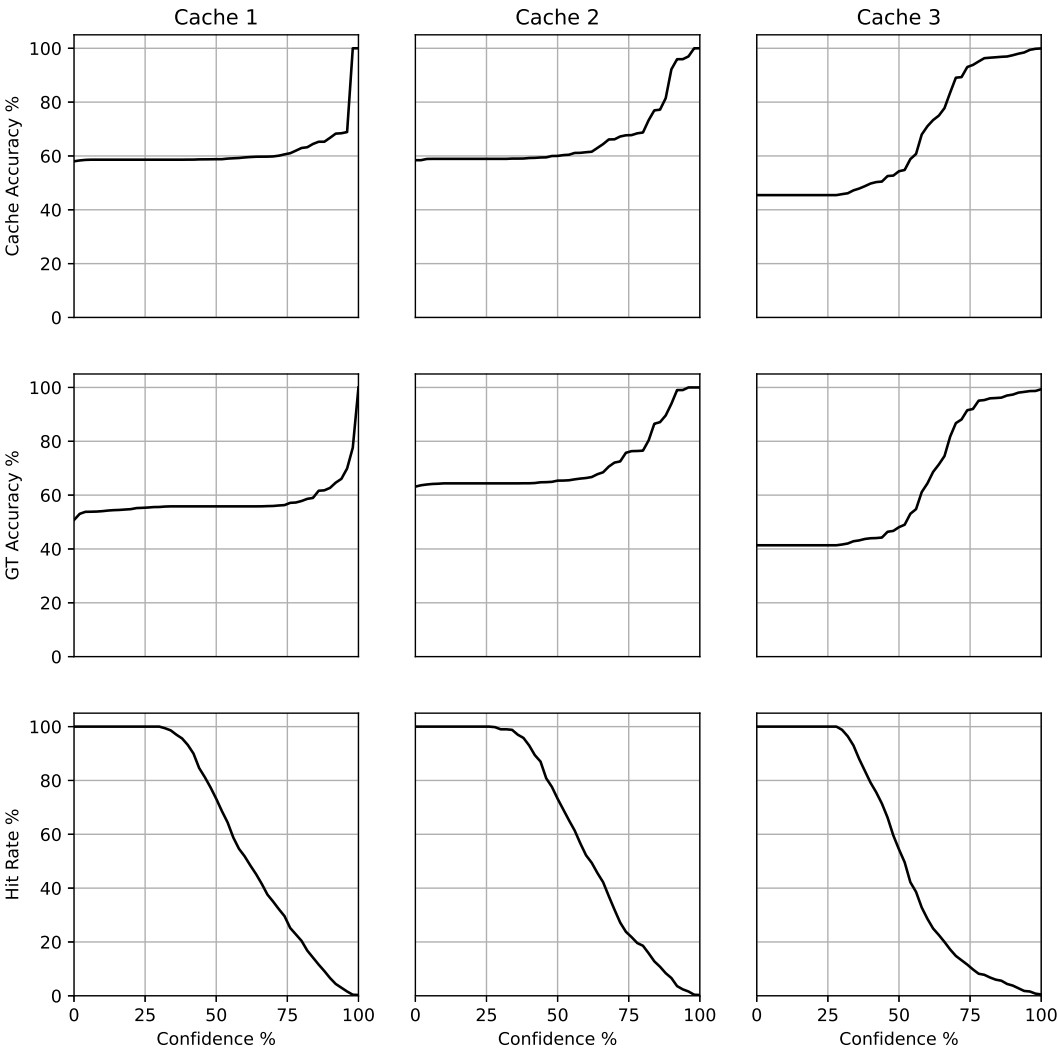

Figure 17: Experiment: DLRM-Criteo

