# OpenReview forum: "SelfXit: An Unsupervised Early Exit Mechanism for Deep Neural Networks"
_TMLR — Accepted by TMLR_

### Review · Reviewer_qSHL · 2024-06-26

**Summary Of Contributions:**

In this paper, the authors propose to use early exits in deep neural networks to return a prediction quickly when these early exits are already able to make an accurate prediction. In addition, the authors propose to update these early exits at runtime, mimicking a caching mechanism.

**Audience:**

Yes

**Broader Impact Concerns:**

-

**Claims And Evidence:**

Yes

**Requested Changes:**

- I feel that the paper should include results on the ImageNet dataset as well. I understand that it is annoying when reviewers ask for additional results on other datasets, especially since the paper already includes experiments on large datasets such as CityScape but the ImageNet dataset is still the most commonly used image classification benchmark and I believe that the large number of classes and the higher resolution makes it a more interesting benchmark than CIFAR10/CIFAR100. In addition, it is likely that follow up work will be evaluated on ImageNet and this would allow those authors to compare to this work.
- The writing could be improved. There are still quite a bit of typo's and numbers in the text have a space after the decimal point.
- Overall, the paper is very verbose which hides a lot of the interesting aspects. I would encourage the authors to focus on what is really new (the online adaptation).

**Strengths And Weaknesses:**

Strengths:
- This paper addresses an interesting topic. Adaptive inference in neural networks is gaining popularity and the idea of using test-time information to improve the efficiency of future predictions is promising.
- While most similar works only report results for image classification, this paper also include object detection and recommendation systems.

Weaknesses:
- I believe the choice of the word "cache" in the title and throughout the paper is misleading. After reading the abstract, I expected that this paper would introduce some sort of key value storage in the model that would allow the model to return a prediction early if the input is similar to a previously processed input, stored in the cache. Instead the authors propose to add trained early exit classifiers. I suppose you could interpret these classifiers as implementing some sort of caching mechanism since they are updated at test-time but I believe that the use of the word "cache" is distracting. Instead, I would encourage the authors to stick to more conventional terminology (e.g. early exit, adaptive computation, ...). I believe this would make it easier for potential readers to find this paper.
- It is not clear to me how the confidence calibration works. I understand the need for calibration but how is this performed without ground truth labels ? I suppose you could use the final model's outputs as ground truth but then you should probably also take this model's confidence into account ?
- The authors put a lot of emphasis on the fact that it is not required to have access to the training data to train the early exits. Does that mean that they are initialized from scratch ? Why not pretrain them on the training data ?
- It is not clear to me how the updating procedure works. Section 3.6 gives two obvious approaches (update every X samples, update the early exits when the model is updated). For a paper that claims "online" in the title, I would have expected a more in depth discussion on the actual online updating procedure. As far as I can see, there is no online aspect in the provided experiments. I would expect at least a graph that shows the accuracy/ runtime of the model as a function of the observed test samples. This way you can see that the model becomes more efficient as more samples have been observed.
- In section 4.2 the authors explain the test procedure. This is not clear to me. For example for the CIFAR10 dataset, the authors state that "we only use the test splits" and "we do not use the labels in these test sets in the training and optimization process". Does that mean that the early exits are trained using test data ? This seems like an obvious opportunity for overfitting. In addition, it is stated that the test data is augmented with flips and rotations which makes it hard to compare to other approaches that just use the conventional test data.

---

### Review · Reviewer_SPN2 · 2024-07-20

**Summary Of Contributions:**

This work presents an approach to accelerate inference for deep neural networks by introducing early exit paths that are trained using test data without requiring labels. The training method works through self-distillation, where the output of each newly added early exit layer mimics the model's output probability distribution (soft labels). The authors also use Neural Architecture Search (NAS) to identify candidate layers for adding early exits, making the approach adaptable to different architectures such as MobileFaceNet, EfficientNet, ResNet18, and ResNet50. Overall, this work extends early exiting approaches to be used in an unsupervised fashion, which brings practical value to applications with limited labeled data.

**Audience:**

Yes

**Broader Impact Concerns:**

Although this is obvious, it may be worth mentioning that deploying this approach in safety-critical applications (e.g., autonomous cars) to improve inference speed may result in accuracy degradation, which could lead to potential accidents.

**Claims And Evidence:**

No

**Requested Changes:**

**A. Recommendation for acceptance (critical comments):**

A.1. The accuracy metrics are not clearly defined and show inconsistent results across the different experiments. For example, the final accuracy of the Base (non-early exit enabled) on CIFAR100-ResNet50 is 78.98% in Table 2, which I would expect to be an approximate upper bound on accuracy based on Table 3 from (Kaya et al., 2019). However, the early exits 1-4 in Figure 2 achieve 100% GT Accuracy at 100% confidence, while the early-exit enabled model achieves roughly 86% accuracy at 95% confidence in Figure 4. In both experiments, I expected these accuracies to be closer and roughly upper bounded by the Base model (78.98%).

**Recommendation:** Similar to the tables of (Kaya et al., 2019), I recommend adding the accuracy of the original model (non-exit enabled) across tables and figures for reference.

A.2. The unsupervised problem setup and evaluation are not clearly defined.
- Given that the training of the early exits is done on the test set, does this mean the model is not deployed yet? Is this training phase on the test set considered a warm-up? Or are predictions being provided to the target application while also training early exits?
- How were other methods trained given that, to my knowledge, they require access to the training data? Were they also trained in an unsupervised way?
- Were other methods evaluated on the entire test set or just the second half of the test data, similar to your method?

**Recommendation:** To address these comments, I recommend adding a formulation of the problem setup in the methodology section since the proposed unsupervised training is part of your contribution and may not be clear to the reader. I also recommend discussing how other methods, which may originally train on labeled data, are adapted to this framework. As a suggestion, this could be under a "Baselines" subsection in Section 4.

A.3. The comparison with other methods is not performed in a standardized way.

- At which confidence % were the results reported in section 4.5.6?

If these were reported at fixed confidence %, this may make the evaluation subjective and could be dataset-specific. While it is great to show a trend of trade-offs between performance and latency, a more standardized approach for benchmarking would be preferable.

**Recommendation:** Since this work is based on (Kaya et al., 2019) and to be consistent with previous works, I recommend showing the comparison results at defined points of inference costs similar to Table 3 of (Kaya et al., 2019).

**B. Comments to strengthen the work (minor comments):**

B.1. Section 3.5 seems to be incomplete or out of context. At the beginning of the section, the discussion appears to be about the implementation of two tasks: image classification and object detection. However, the rest of the section only covers image classification for pedestrian detection. Then, at the end of the section, a recommendation system task is introduced abruptly without sufficient context. I recommend revising Section 3.5 to clarify its objective.

B.2. Writing clarity: I highly recommend revising the document thoroughly in terms of language and grammar. For example:
- There are several mentions of the term “a early exit”, which should be “an early exit”.
- The phrase “acceleration inference” in Section 2.1 should be “accelerating inference”.

B.3. Terminology consistency: There are a few mentions of “mean precision” and “final precision”. It seems precision and accuracy are used interchangeably in the document, even though they are different metrics. Do these references actually indicate the precision metric? If so, I suggest defining them in the metrics.

**Strengths And Weaknesses:**

**Strengths:**
1. The proposed method works in an unsupervised manner without requiring labeled data, making it useful for applications with limited labeled data. The authors also use Neural Architecture Search (NAS) to dynamically add early exits across different architectures, demonstrating great flexibility and adaptability.
2. The related works section is comprehensive and clearly differentiates this work’s contribution from the existing literature.
3. The proposed approach shows significant speed-ups in inference compared to other methods at the cost of a slight drop in accuracy. This can be useful in applications where the inference cost is prioritized over accuracy.

**Weaknesses:**

1. The accuracy metrics are not clearly defined and show inconsistent results across the different experiments. (Refer to Requested Changes A.1 for details)
2. The unsupervised problem setup and evaluation are not clearly defined. (Refer to Requested Changes A.2 for details)
3. The comparison with other methods is not performed in a standardized way. (Refer to Requested Changes A.3 for details)

**Disclaimer:**
I skimmed the math related to Equation 3 and am not able to evaluate its theoretical correctness.

---

> ### Author Response · Authors · 2024-08-12
> **Manuscript Revision for Review**
>
> Thank you for your constructive feedback. I have revised the manuscript to address your considerations and have submitted it for your review.

---

> ### Comment · Reviewer_SPN2 · 2024-08-13
>
> Thank you for addressing my comments. I will respond to each main point below:
>
> **A.1.**  I appreciate the clarification. To enhance understanding for readers, I suggest incorporating the explanations you provided for the items below into the paper:
> - The discrepancy between layer-by-layer accuracy (early exit accuracy?) and final accuracy.
> - The fact that the upper bound (Base) does not apply to other methods.
>
> **A.2.** Thanks for addressing the baseline request. However, there is still an unaddressed part, which is defining the formulation of the problem setup. For a more concrete example, since you are extending the work of (Kaya et al., 2019), refer to their 'Setting' subsection (first paragraph of Section 3). It is essential to formally define your unsupervised setting so that it can be easily compared with other settings.
>
> **A.3.** I believe there is a misunderstanding regarding the original request. The request had two parts:
>
> (1) The results in Section 4.6.6 reference some values for the figures at an undefined confidence interval. For example, the statement "we achieved a substantial 52.2% and 32.4% reduction in latencies, with a modest 6.9% and 1.6% decrease in mean accuracy when compared to the BranchyNet and Gati method" doesn't define the confidence interval at which the observation was made from Figure 3. If the same confidence interval is used for all observations, this can be simply addressed by stating the confidence value at the beginning of Section 4.6.6.
>
> (2) The current presentation of the results in Figures 3 and 4 compares methods across three metrics, making it difficult to compare them directly. Table 3 of (Kaya et al., 2019) reports accuracy at inference costs of ≤25%, ≤50%, and ≤75%, allowing for a direct comparison of cost to accuracy. In contrast, reporting accuracy versus the confidence interval does not directly reflect the trade-off between inference cost and accuracy. Additionally, the table in Appendix D only reports Early Exit Enabled and omits BranchyNet and Gati. I believe generating the table I'm requesting does not require running new experiments; you can simply extract the inference costs of ≤25%, ≤50%, and ≤75% from Figures 3 and 4.

---

> > ### Comment · Reviewer_SPN2 · 2024-09-01
> > **Question about the evaluation/training conditions**
> >
> > Dear authors,
> >
> > Thank you for providing your revision.
> >
> > I still have concerns about the comparisons with other methods. The new table (Table 7) indicates that your proposed method only outperforms others when using a larger backbone (ResNet50 compared to ResNet18) and a larger dataset (CIFAR-100 compared to CIFAR-10). However, this performance gain is also observed in the Base model, raising questions about whether your early exit approach is actually superior or if differing training conditions are affecting the results, making the comparison potentially unfair.
> >
> > My main concern is understanding why this behavior occurs. In Figures 3 and 4, the Base model for CIFAR-10 with ResNet18 performs worse than BranchyNet and Gati at maximum confidence, possibly due to the "overthinking" phenomenon. However, I am unclear why the Base for CIFAR-100 with ResNet50 outperforms these methods. My understanding is that early exits can improve performance, but they shouldn’t cause significant degradation when confidence is maxed out. Could there be other factors contributing to the observed advantage of the Base and, consequently, your method on CIFAR-100 with ResNet50? Any insights or explanations would be helpful.
> >
> > Additionally, based on the comment (Review 1-2), is the Base model pre-trained on the same target dataset (e.g., CIFAR-100 with ResNet50 pre-trained on CIFAR-100)? If so, the Base model, using "supervised training", should have similar training conditions, so the roughly 5% discrepancy in accuracy between Gati, BranchyNet, and the Base on CIFAR-100 with ResNet50 is unexpected.
> >
> > I would appreciate it if you could provide more context on your evaluation and training conditions, as well as an explanation for this behavior.

---

> > > ### Author Response · Authors · 2024-09-04
> > > **Response on Model Training and Evaluation Setup**
> > >
> > > Thank you for your quick response and for the detailed and insightful comments.
> > >
> > > We would like to address all concerns and provide further clarification.
> > >
> > > First, let us explain the base model, training, and evaluation setup used in our experiments. We define the pre-trained model (or the model already trained on the training dataset) as the Base model. We observed, both in our work and in other baselines, that there is a notable trade-off between accuracy and latency. Achieving better inference times can sometimes lead to a slight decrease in accuracy, though this impact is generally minimal.
> > >
> > > In our comparison, we noted that Gati used the validation dataset to train cache layers. To ensure a fair comparison, we used the same data for training the layers across all methods. Specifically, BranchyNet employs both training and validation data, while Gati uses a pre-trained method similar to ours, with BranchyNet training from scratch. However, both methods do not freeze their base models while training layers, which is a crucial difference highlighted in our experiments (accessible in an anonymous GitHub repository).
> > >
> > > Initially, we did not expect to see better results than the Base model in Gati and BranchyNet, especially since overthinking does not occur in their methods, as they have mentioned. However, as shown in Figure 3 (CIFAR 10), we observed this, which was surprising. Similarly, we did not anticipate the worse performance in high-confidence scenarios from those methods, as shown in Figure 4 (CIFAR 100). After re-running the experiments multiple times and observing similar behaviors, we identified two major factors contributing to these observations:
> > >
> > > First, in our method, we completely freeze the base model, preventing any changes during the training of our early-exit layers. This approach may lead to better results within a limited training time, especially in more dense and complex models, where other methods might require significantly more time to train effectively. Second, we train layers across all methods using a portion of the test dataset (the half not used for evaluation) to ensure a fair comparison. In our unsupervised method, we aim to make early-exit results resemble the trained model rather than converging to the ground truth. Consequently, our upper bound is consistently the Base model, while other methods may exhibit varying test accuracies depending on additional training, which is expected.
> > >
> > > It is important to note that in simpler models and datasets, those methods demonstrated better results than our base model. However, within the same limited training epochs applied to all methods, we consistently observe that our upper bound is the Base model due to the unsupervised nature of our training, which fine-tunes early-exit layers to align closely with the Base model. Additionally, regarding performance at high confidence thresholds, it was observed that these methods maintain a very high hit rate (even over 97% confidence). This results in behavior that diverges from the base model, even at high confidence levels. For example, 100% confidence in our work may lead to a zero hit rate, while they show a 10-15% hit rate in such cases. As mentioned earlier, our method appears to be more adjustable in this regard.

---

### Review · Reviewer_t26e · 2024-08-16

**Summary Of Contributions:**

This paper proposes an end-to-end automated early exiting solution to improve the performance of DNN-based services in terms of computational complexity and inference latency. The paper methodology involves - (a) Identify the candidate layers to be early exit (b) Build an early exit model for each candidate (c) Assign confidence thresholds to built models to determine early exit hits (d) Evaluate and optimize the early exit-enabled model (e) Early-Exit Optimization Implementation (f) Update and maintenance early exit models.

**Audience:**

Yes

**Broader Impact Concerns:**

None.

**Claims And Evidence:**

No

**Requested Changes:**

The critical requirements include a through investigation of training and inference time compute benefits, and details for  hardware/software stack is required. The latency increases with increasing the batchsize (Table 6) is also a negative point, given modern GPUs which are fast and efficient in large batch processing. I also highly encourage the authors to explain their key method with a good diagram to make it easy to understand, at this point it requires a lot of effort to understand it to understand the nuances. Also, additional comments related to adaptation for modern architectures with stack of transformer will be helpful given the candidate layer selection is heuristic based.

**Strengths And Weaknesses:**

Strengths:

1. The draft is well written with proper inclusion of past related work to easily follow the related concepts for new readers.
2. The idea is developed in unsupervised settings, allowing early exit models do not need access to training data and perform solely based on the incoming data at run-time.
3. During the training an early exit model, the proposed work freeze the the model backbone, to ensure that the training process does
not modify any parameter not belonging to the current early exit model; which encourages adoptability.
4. The authors extends their proposed work to evaluating a critical real-world scenario - pedestrian detection in urban environments.


Weakness:

While the paper is a interesting read, I have several concerns with the algorithm/method developed given its complexity.

1. In candidate layer selection step (section 3.1), the author mentions several static rules to select candidate layers (eg. last layers not useful, layers with parallel connections are not useful). I find this static rules difficult to generalize across numerous existing backbone architecture. How should one decide how many last layers to discard? The method is not generalizable for recent popular complex architectures like MoEs where there a multiple parallel experts?

2. Does the authors have thought about using these early exit approaches for modern LLMs where early exits would add a lot of value given gigantic memory cost? Recently some work find like SkipDecode (https://arxiv.org/abs/2307.02628), ShortGPT(https://arxiv.org/pdf/2403.03853), FFN-SkipLLM (https://arxiv.org/abs/2404.03865) explores layer-skipping or early-exit strategies. What are authors thought about these techniques given their simplicity in identifying comparatively less useful layers and discarding them in limited compute settings?

3. In section 3.2.1, the author briefly discuss the complexity of task (CIFAR10 vs CIFAR100) in deciding the amount the early-exit model capacity. I see further potential here to explore a more task-difficulty specific evaluation to see how it relate with the speedup achieved for a easy task vs difficult task?

4. Comparison with a dense baseline with similar FLOP/compute count to validate the performance difference of the early exit model with a model with similar compute cost is required.

5. The well-advertised benefit of the work is speedup, however a clean speedup/latency evaluation settings is missing. How is the speed up measured specifically across CPUs and GPUs? What's the hardware/software stack? A fine-grained evaluation of compute/latency/memory for backbone, early-exit modules is a must? The authors nee to also provide an idea of compute additionally involved in training these early exit models, NAS, thresholding.

6. I am also curious about the thresholding, and the expense of update and maintenance of early exit models. A ablation of these thresholding is required to show the impact it can have.

---

> ### Author Response · Authors · 2024-08-21
> **Request for Clarifications**
>
> Dear Reviewer,
>
> Thank you for your valuable feedback.
>
> While addressing your concerns, we encountered two questions that we believe will help us better address your points:
>
> 1. **Regarding the diagram you requested,**
> We have illustrated our system architecture in Figure 1 and Figure 5 (in the appendix) in attempt to clearly demonstrate our methodology. We kindly ask if more specific details can be provided to improve the clarity of our architecture diagrams.
> 2. **Concerning Weakness 4, highlighted in the requested changes,**
> We kindly ask the reviewer to further clarify this request change. To our best understanding, the requested change suggests that we implement a different ResNet model with similar FLOPS to our early exit method during inference. If our understanding is correct, the requested change may not be feasible, as there are no standard ResNet models other than 18, 34, 50, 101, and 150. Similarly, for other models and applications, we have used standard pre-trained models, and our objective was on improving latency with minimal accuracy loss. Creating another model with identical FLOPS might be orthogonal to our objective.

---

### Decision · Action_Editor_DXxw · 2024-09-23

**Recommendation:** Accept with minor revision

**Comment:**

Two reviewers lean toward rejection (qSHL, SPN2) while one reviewer leans toward acceptance (t26e). The vote for acceptance by t26e is still qualified by concerns about the thresholding and the computational expense of the early exit models, but they side with leaning accept, because their concerns have been addressed and the submission can be understood by the reader and is therefore informative. The votes for rejection both agree that the work has an interesting topic of great relevance to the community, and that there are valuable insights for early exiting especially for practitioners, but on the balance they are concerned with (1) the clarity of the setting evaluation to identify what is learned from this work and (2) the comparability of experiments between the proposed method and the comparison methods of BranchyNet and GATI. qSHL also raised an issue about novelty, but this has been factored out of the decision given TMLR's mission to consider claims-evidence and audience, and an issue about missing ImageNet, but ImageNet results are partially provided in the revision.

The concerns (1) and (2) about clarity and comparability are serious and would require rejection if unresolved. However, (1) has been improved by the revision and discussion, and can improve further with a last revision. (2) is not relevant to the internal claims of this work about the effect of the proposed method vs. the base model but the issue is key to comparisons and the claims in the abstract about latency and accuracy. Reviewer SPN2 has made multiple important and constructive points on this and the authors have replied. The submission is not yet fully revised to incorporate this feedback, but can be (and must be or else the claims and evidence are not fully accurate and concrete).

The action editor decides on acceptance with minor revisions but must note that the revisions while minor in scope are major in importance and required for acceptance. The reasons for acceptance are:

- The results on the proposed early existing model are informative whether both positive or negative. For instance, the pairing of Tables 5 & 6 shows both the potential for time efficiency and the pitfall of batching with current hardware. Likewise there is a plus and a minus for memory efficiency, in that exiting could allow memory to be reclaimed with dynamic allocation but the early-exit architecture requires more parameters and may need more total memory when executing all exits. Explaining and measuring these concerns is valuable especially for the multiple measures of FLOPs, memory, and latency as done here. In this work the results make clear that the computational efficiency is in time (for suitable batch sizes), and not memory, since more overall memory is needed.
- Following the discussion by SPN2, the experiments with BranchyNet and GATI are fair modulo the differences in setting, even if they are not as complete as could be. The results do control for the use of data and amount of training even if there are differences in pre-training and fixing parameters. In particular, there could be an ablation and analysis of whether to freeze the base model or not. However, without it the results are still true as a holistic evaluation of the proposed method vs. BranchyNet and GATI in all their details.
- The introduction and related work can guide those interested in efficient inference for image classification and inspire more work. This is worth noting because this work provides pointers to inference optimization more broadly, early exiting for different purposes, and distillation/self-distillation for training models on model predictions.

The requested revisions distill the official recommendations by reviewers and the concerns summarized by the action editor in the decision:

1. Scope the abstract and summary of contributions to limit the claims to deep convolutional networks for vision. This addresses the feedback and decision recommendations from reviewers about different model types like transformers and domains like text and is preferable to the general discussion of deep networks.
2. Resolve the lack of clarity around the comparability of the compared methods. Make sure to incorporate the discussion of how training and evaluation are related (including this comment https://openreview.net/forum?id=h4rUKKfl5S&noteId=7fcilGsOub) for the proposed method, BranchyNet, and GATI.
3. Discuss this reference on unsupervised early exiting in NLP "Unsupervised Early Exit in DNNs with Multiple Exits" by Narayan et al. 2022 which was provided by SPN2 during the recommendation phase. Note that the domain and methodology differ and do not invalidate the claims and evidence of this submission.
4. Of course the draft revision with comments to the reviewers and the like must be edited into a final revision.

The action editor offers this miscellaneous feedback:

- As a suggestion for terminology, "distillation" or "self-training" may be more accessible and avoid confusion as alternatives to "unsupervised learning" because some readers may expect the training to be a function of the input itself, like unsupervised reconstruction losses or self-supervised auxiliary losses, rather than the training to be a function of the base model predictions. Of course this is up to the authors!
- Sec. 3.6. It is confusing to talk about updates and maintenance if these are not done in this work. Consider removing this section and moving the content on evaluation stages to the following Sec. 4 on evaluation. These thoughts on model maintenance could be added to a discussion at the end such as in Sec. 4.7 to group together discussion, limitations, and future work.
- Sec. 4.5. Add parenthetical citations for the baselines for ease of reference within the section.
- pg. 7. ImageNet is missing from the list of datasets in the empirical contributions.
- pg. 8. Possible typo: "utilize their old observation" for "utilize their own observation"? If not, then consider rephrasing for clarity anyway.
- pg. 8. Typo: "overthinkingproblem"
- pg. 25. There is a stray fragment "They must be comprised of accuracy drop data" before Table 3.
- pg. 30. Major typo: "goal of our proposed approach is to increase the inference time" when it should be "decrease the inference time".
- pg. 30. Typo: "Appendix .2."
- The capitalization of GATI is inconsistent. Please use GATI to respect the capitalization in the original paper.

The action editor thanks the authors, for submitting to TMLR and for engaging in the discussion and revision, and thanks the reviewers, for the service to TMLR and for providing their expertise and constructive feedback.

**Audience:**

All three reviewers agree that there is an audience. This work is informative for researchers in computer vision working on early exiting specifically and efficient inference more generally. This work may also apply to other architectures and modalities, but more work is needed to show this, and the text appropriately acknowledges that the scope is limited to visual recognition tasks and residual network backbones.

The action editor agrees with the reviewers and confirms that there is an audience. This dimension has improved since the last submission, with the clarification of the addition of a broader set of results including ImageNet, and so it is therefore no longer borderline.

**Claims And Evidence:**

Two reviewers agrees that the claims match the evidence (t26e, qSHL) and one reviewer disagrees (SPN2). For t26e the claims and evidence are supported when the provided clarifications and discussion of limitation are included. For SPN2 the key issues are identifying the setting for unsupervised early-exiting and fair comparison with existing methods. After discussion these issues are not fully resolved, although more details are provided by the authors, and at least some potential confounds (such as the choice of data) are controlled for.

The action editor finds that the claims and evidence partially agree. The "internal" results of the base modela and proposed early-exit model agree with the contributions, but the "external" results of comparison among methods are not currently adequately explained and scoped. See the recommendation and comment for more detail on this and the necessary changes.